# HIV-1 integrase tetramers are the antiviral target of pyridine-based allosteric integrase inhibitors

Pratibha C Koneru[1], Ashwanth C Francis[2], Nanjie Deng[3], Stephanie V Rebensburg[1], Ashley C Hoyte[1], Jared Lindenberger[1], Daniel Adu-Ampratwum[4], Ross C Larue[4], Michael F Wempe[5], Alan N Engelman[6,7], Dmitry Lyumkis[8], James R Fuchs[4], Ronald M Levy[9], Gregory B Melikyan[2], Mamuka Kvaratskhelia[1]*

[1]Division of Infectious Diseases, School of Medicine, University of Colorado, Aurora, United States; [2]Division of Infectious Diseases, Department of Pediatrics, Emory University, Atlanta, United States; [3]Department of Chemistry and Physical Sciences, Pace University, New York, United States; [4]College of Pharmacy, The Ohio State University, Columbus, United States; [5]Skaggs School of Pharmacy and Pharmaceutical Sciences, University of Colorado Denver, Aurora, United States; [6]Department of Cancer Immunology and Virology, Dana-Farber Cancer Institute, Boston, United States; [7]Department of Medicine, Harvard Medical School, Boston, United States; [8]Laboratory of Genetics, The Salk Institute for Biological Studies, La Jolla, United States; [9]Department of Chemistry, Temple University, Philadelphia, United States

*For correspondence:
mamuka.kvaratskhelia@ucdenver.edu

**Abstract** Allosteric HIV-1 integrase (IN) inhibitors (ALLINIs) are a promising new class of antiretroviral agents that disrupt proper viral maturation by inducing hyper-multimerization of IN. Here we show that lead pyridine-based ALLINI KF116 exhibits striking selectivity for IN tetramers versus lower order protein oligomers. IN structural features that are essential for its functional tetramerization and HIV-1 replication are also critically important for KF116 mediated higher-order IN multimerization. Live cell imaging of single viral particles revealed that KF116 treatment during virion production compromises the tight association of IN with capsid cores during subsequent infection of target cells. We have synthesized the highly active (-)-KF116 enantiomer, which displayed $EC_{50}$ of ~7 nM against wild type HIV-1 and ~10 fold higher, sub-nM activity against a clinically relevant dolutegravir resistant mutant virus suggesting potential clinical benefits for complementing dolutegravir therapy with pyridine-based ALLINIs.

## Introduction

Multifunctional HIV-1 integrase (IN) is an important therapeutic target. HIV-1 IN strand transfer inhibitors (INSTIs), which have become a first-line therapy to treat HIV-1 infected patients, block the catalytic function of the viral protein during the early phase of infection and thus prevent integration of viral cDNA into human chromosomes (*Hazuda, 2012*; *Hazuda et al., 2000*; *McColl and Chen, 2010*). Clinical applications of first generation INSTIs raltegravir (RAL) and elvitegravir (EVG) resulted in evolution of drug resistant phenotypes containing IN substitutions in the vicinity of the inhibitor binding sites (*Cooper et al., 2008*; *Garrido et al., 2012*). Second generation dolutegravir (DTG), bictegravir (BIC) and investigational drug cabotegravir (CAB) exhibit significantly higher genetic pressure for the evolution of drug resistance and yield complex resistance pathways with IN

**eLife digest** HIV-1 inserts its genetic code into human genomes, turning healthy cells into virus factories. To do this, the virus uses an enzyme called integrase. Front-line treatments against HIV-1 called "integrase strand-transfer inhibitors" stop this enzyme from working. These inhibitors have helped to revolutionize the treatment of HIV/AIDS by protecting the cells from new infections. But, the emergence of drug resistance remains a serious problem. As the virus evolves, it changes the shape of its integrase protein, substantially reducing the effectiveness of the current therapies. One way to overcome this problem is to develop other therapies that can kill the drug resistant viruses by targeting different parts of the integrase protein. It should be much harder for the virus to evolve the right combination of changes to escape two or more treatments at once.

A promising class of new compounds are "allosteric integrase inhibitors". These chemical compounds target a part of the integrase enzyme that the other treatments do not yet reach. Rather than stopping the integrase enzyme from inserting the viral code into the human genome, the new inhibitors make integrase proteins clump together and prevent the formation of infectious viruses. At the moment, these compounds are still experimental. Before they are ready for use in people, researchers need to better understand how they work, and there are several open questions to answer. Integrase proteins work in groups of four and it is not clear how the new compounds make the integrases form large clumps, or what this does to the virus. Understanding this should allow scientists to develop improved versions of the drugs.

To answer these questions, Koneru et al. first examined two of the new compounds. A combination of molecular analysis and computer modelling revealed how they work. The compounds link many separate groups of four integrases with each other to form larger and larger clumps, essentially a snowball effect. Live images of infected cells showed that the clumps of integrase get stuck outside of the virus's protective casing. This leaves them exposed, allowing the cell to destroy the integrase enzymes.

Koneru et al. also made a new compound, called (-)-KF116. Not only was this compound able to tackle normal HIV-1, it could block viruses resistant to the other type of integrase treatment. In fact, in laboratory tests, it was 10 times more powerful against these resistant viruses. Together, these findings help to explain how allosteric integrase inhibitors work, taking scientists a step closer to bringing them into the clinic. In the future, new versions of the compounds, like (-)-KF116, could help to tackle drug resistance in HIV-1.

substitutions in both the proximity of and significantly distanced from inhibitor binding sites (*Cahn et al., 2013*; *Eron et al., 2013*; *Malet et al., 2018*; *Margolis et al., 2015*; *Smith et al., 2018*; *Tsiang et al., 2016*).

More recently, allosteric HIV-1 integrase (IN) inhibitors (ALLINIs; also referred to as non-catalytic site integrase inhibitors (NCINIs); LEDGINs or INLAIs), which are mechanistically distinct from INSTIs have been developed (*Christ et al., 2010*; *Fader et al., 2014b*; *Le Rouzic et al., 2013*; *Sharma et al., 2014*; *Tsiang et al., 2012*). Unlike INSTIs, ALLINIs are highly potent during virion maturation. They induce higher-order IN multimerization in virions and consequently inhibit IN-RNA interactions, which in turn yields eccentric noninfectious virions with the ribonucleoprotein complexes (RNPs) mislocalized outside of the translucent capsid (CA) cores (*Balakrishnan et al., 2013*; *Desimmie et al., 2013*; *Fontana et al., 2015*; *Gupta et al., 2014*; *Jurado et al., 2013*; *Kessl et al., 2016*; *van Bel et al., 2014*). ALLINIs also display a secondary, albeit significantly reduced, activity during early steps of HIV-1 infection where these compounds interfere with HIV-1 IN binding to its cognate cellular cofactor LEDGF/p75 (*Christ and Debyser, 2013*; *Christ et al., 2012*; *Christ et al., 2010*; *Kessl et al., 2012*; *Le Rouzic et al., 2013*; *Sharma et al., 2014*; *Tsiang et al., 2012*).

The key pharmacophore in all potent ALLINIs includes both a carboxylic acid and *tert*-butoxy group, which extend from a core aromatic ring system (*Figure 1A*) (*Christ et al., 2010*; *Jurado and Engelman, 2013*; *Sharma et al., 2014*; *Tsiang et al., 2012*). The original series of compounds contained a quinoline core, exhibited relatively modest potency, and comparatively low genetic barrier to drug resistance (*Christ et al., 2010*; *Fader et al., 2014b*; *Feng et al., 2013*; *Tsiang et al., 2012*). Follow up SAR studies have extended in two directions: i) efforts that retain the quinoline core while

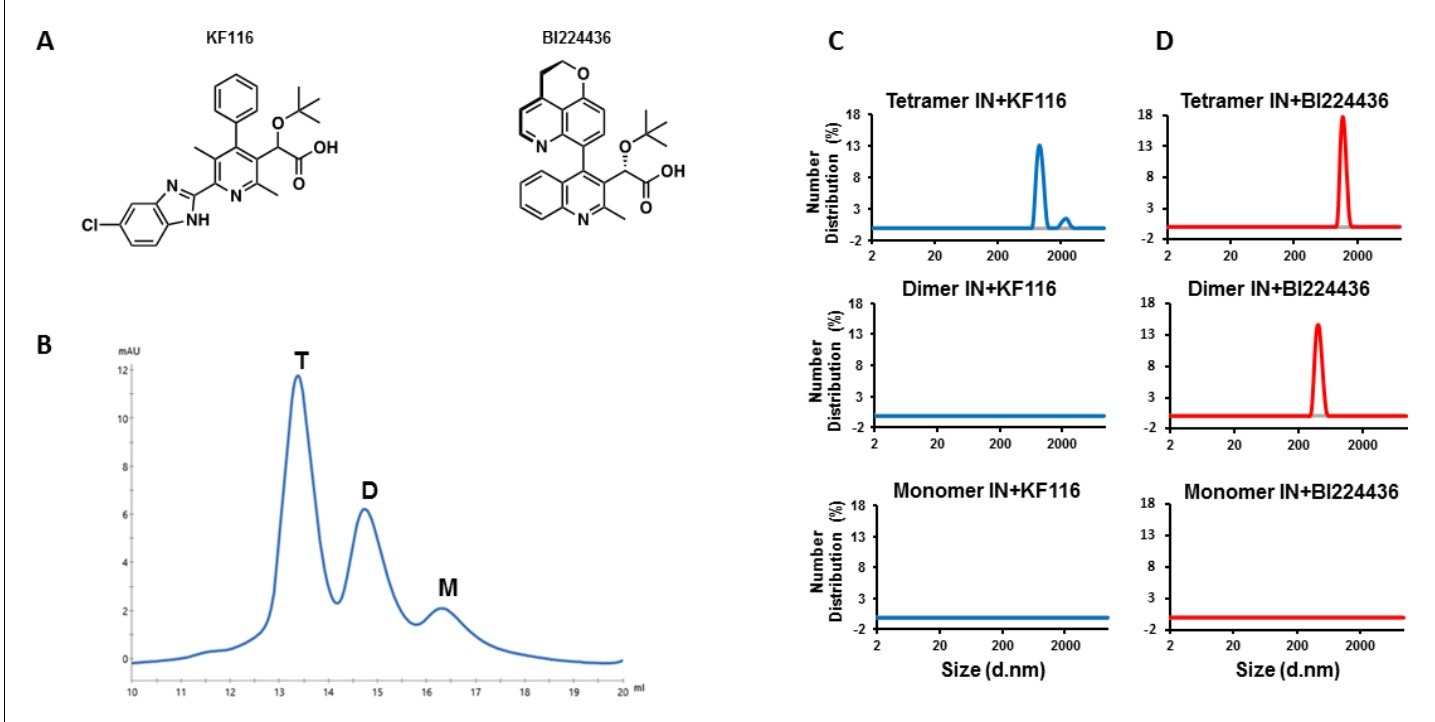

**Figure 1.** IN tetramers are preferentially targeted by pyridine and quinoline based ALLINIs. (A) Chemical structures of pyridine-based KF116 and quinoline-based BI224436. (B) SEC based separation of IN tetramer(T), dimer(D) and monomer(M) fractions. C and D, DLS analysis of 200 nM IN fractions in the presence of 1 μM KF116 (blue lines, (C) or BI224436 (red lines, (D). DMSO controls for each DLS experiment (C and D) are shown in gray. Representative results of three independent experiments at 30 mins time point are shown.

The online version of this article includes the following figure supplement(s) for figure 1:

**Figure supplement 1.** Pyridine and quinoline based ALLINIs preferentially target IN tetramers.

varying the substituted groups have resulted in highly potent inhibitors such as BI224436 with antiviral $EC_{50}$ of ~14 nM (*Fader et al., 2014b*) (*Figure 1A*); ii) other studies successfully explored different core ring structures (*Demeulemeester et al., 2014*; *Fader et al., 2016*; *Patel et al., 2016a*; *Patel et al., 2016b*; *Sharma et al., 2014*) to also synthesize highly potent ALLINIs. We and others have rationally developed pyridine-based compounds that exhibit markedly improved antiviral activities and significantly enhanced genetic pressure for the evolution of resistance compared with archetypal quinoline-based compounds (*Fader et al., 2016*; *Hoyte et al., 2017*; *Sharma et al., 2014*). For example, our lead racemic pyridine-based compound, KF116 (*Figure 1A*), exhibited ~24 nM antiviral activity and remained fully potent with respect to A128T IN mutant virus that confers significant resistance to archetypal quinoline-based ALLINIs. Instead, triple IN substitutions (T124N/V165I/ T174I), which substantially compromise viral replication, were needed to confer effective resistance to KF116 (*Hoyte et al., 2017*; *Sharma et al., 2014*). Furthermore, extensive in vitro absorption, distribution, metabolism, and excretion (ADME) and animal pharmacokinetic studies of quinoline- and pyridine-based inhibitors performed by Boehringer Ingelheim (BI) have indicated the superiority of the pyridine-based compounds as 'a clinically viable starting point' (*Fader et al., 2016*; *Fader et al., 2014a*; *Fenwick et al., 2014*).

IN is comprised of three structurally distinct domains: N-terminal domain (NTD), catalytic core domain (CCD) and C-terminal domain (CTD) (*Chiu and Davies, 2004*; *Engelman and Cherepanov, 2014*). Biochemical and x-ray crystallography experiments with individual domains revealed that ALLINIs selectively bind to a v-shaped pocket located at the CCD dimer interface (*Fader et al., 2014b*; *Feng et al., 2013*; *Sharma et al., 2014*). Mechanistic studies with various ALLINI core ring structures have shown that the cores do not directly interact with IN. Instead, they enable the conserved pharmacophore carboxylic acid and *tert*-butoxy groups to accurately position within the

v-shaped pocket allowing hydrogen bonding with backbone amides of IN residues Glu-170 and His-171 and the side chain of Thr-174, respectively.

Mass spectrometry-based protein foot-printing experiments (*Feng et al., 2016*; *Shkriabai et al., 2014*) with full-length IN identified that in addition to the CCD, the CTD plays a critical role for inhibitor induced hyper-multimerization of IN. In turn, these findings enabled molecular modeling studies, which suggested that ALLINIs directly promote inter-subunit interactions between the CCD dimer and the CTD of another IN dimer (*Deng et al., 2016*). Furthermore, free energy calculations revealed that in the absence of the inhibitor the v-shaped binding cavity on the CCD dimer is occupied with thermodynamically unstable water, which minimizes potential CCD-CTD interactions. In contrast, ALLINI binding to the CCD dimer effectively fills the v-shaped pocket by displacing the water molecules and thus markedly enhances the binding interface for incoming CTD.

More recently, the crystal structure of full-length HIV-1 IN Y15A/F185H in complex with quinoline-based ALLINI GSK1264 was reported (*Gupta et al., 2016*), which confirmed both the earlier biochemical results and predicted model of a multimer interface involving CCD-inhibitor-CTD interactions. The crystal structure revealed two IN Y15A/F185H dimers bridged by two quinoline-based GSK1264 inhibitors through head (CCD-CCD dimer) to tail (CTD of another dimer) interactions. However, technical limitations of the crystallographic experiments (*Gupta et al., 2016*) have left the following important questions unanswered: (i) what is the relevant oligomeric state of WT IN in the context of ALLINI induced higher-order multimerization? The crystal structure used mutant IN with Y15A and F185H substitutions, which alters the in vitro multimeric state of IN (IN Y15A/F185H and WT IN are predominantly dimer and tetramer, respectively). The Y15A substitution impairs HIV-1 replication (*Takahata et al., 2017*), however Y15A/F185H substitutions were critical to slow down the inhibitor induced IN multimerization allowing for the capture of smaller complexes amenable to x-ray crystallography. Therefore, it is unclear which oligomeric state(s) of WT IN is(are) the authentic target for these inhibitors. (ii) Is there a role for the NTD in the inhibitor induced higher-order IN multimerization? The NTD structure could not be resolved in the x-ray structure of the IN Y15A/F185H + ALLINI GSK1264 complex. Therefore, a role for the NTD in inhibitor induced higher-order IN multimerization remains to be elucidated. (iii) How do distinct ALLINI scaffolds induce higher-order IN multimerization? Pyridine-based compounds are structurally distinct from their quinoline-based counterparts. Therefore, it is critically important to understand similarities and differences for how pyridine- vs quinoline-based ALLINIs induce hyper-multimerization of WT IN.

To address the above questions, we have investigated key structural determinants of WT IN for ALLINI induced higher-order multimerization. Our experiments focused on analyzing the mode of action of lead pyridine-based ALLINI KF116, while using the highly potent quinoline-based BI224436 in side-by-side comparisons (*Figure 1A*). We demonstrate that KF116 selectively and BI224436 preferentially target WT IN tetramers. Furthermore, we have synthesized the highly active (-)-KF116 enantiomer, which exhibits substantially enhanced, sub-nM potency with respect to the therapeutically important DTG resistant virus containing IN N155H/K156N/K211R/E212T substitutions (*Malet et al., 2018*). These findings suggest potential clinical benefits for combining KF116 and DTG therapies to limit HIV-1 options for developing drug resistant variants in patients.

## Results

### KF116 specifically and BI224436 preferentially targets full-length WT IN tetramers

In vitro preparations of full-length WT HIV-1 IN yield a mixture of tetrameric, dimeric and monomeric forms. To examine which of these forms are targeted by ALLINIs, we have separated different oligomeric forms of WT IN by size exclusion chromatography (SEC) (*Figure 1B*). To prevent re-equilibration of separated species, equimolar concentrations of tetramers, dimers and monomers were immediately incubated with KF116 and formation of the inhibitor induced higher-order IN multimers were monitored by dynamic light scattering (DLS). DLS is an optical method for studying the diffusion behavior of macromolecules in solution (*Stetefeld et al., 2016*). While unliganded IN does not yield any detectable signal due to relatively small sizes of fully soluble monomeric, dimeric and tetrameric forms, ALLINI induced higher-order IN oligomers are readily detected by DLS (*Sharma et al., 2014*). The results in *Figure 1C* show that KF116 specifically induced higher-order oligomerization

of tetramers but not dimers or monomers. More detailed kinetic analysis (*Figure 1—figure supplement 1A*) revealed that within 1 min after addition of KF116 to IN tetramers, higher-order protein multimers with particle diameter sizes of ~200 nm were formed. Particle diameter sizes increased further to ~1,000 nm in a time dependent manner indicating an equilibrium shift towards higher-order multimers (*Figure 1—figure supplement 1A*). In sharp contrast, the inhibitor failed to alter the multimeric form of dimers or monomers even after a 60 min incubation (*Figure 1C* and *Figure 1—figure supplement 1B*). The absence of any higher-order multimerization of these preparations provides corollary evidence that the separated lower oligomeric forms of IN do not detectably re-equilibrate into tetramers over the incubation time.

In parallel reactions, we have analyzed the quinoline-based BI224436, which exhibited a broader specificity for tetramers and dimers but did not have any effects on IN monomers (*Figure 1D* and *Figure 1—figure supplement 1C–D*). Addition of BI224436 to IN tetramers yielded higher-order multimers within 1 min (*Figure 1—figure supplement 1C*), whereas a longer incubation time (at least 30 min) for BI224436 +IN dimers was needed to detect higher-order IN multimers (*Figure 1—figure supplement 1D*). Collectively, these findings indicate that KF116 selectively and BI224436 preferentially targets IN tetramers.

## KF116 and BI224436 exhibit differential stoichiometry for higher-order IN multimerization

The crystal structure of IN(Y15A/F185H)+GSK1264 (*Gupta et al., 2016*) revealed that two IN(Y15A/F185H) dimers are bridged by two quinoline-based inhibitors (i.e., 2:2 stoichiometry). Here we determined the stoichiometry for KF116 and BI224436 interactions with WT IN. For this, we have quantified dose dependent effects of ALLINI induced IN aggregation (*Figure 2* and *Figure 2—figure supplement 1*). The results in *Figure 2* indicate a 2:4 ratio for KF116:IN suggesting that two molecules of KF116 bridge two IN tetramers. In contrast, BI224436:IN interactions exhibited a 2:2 or 4:4

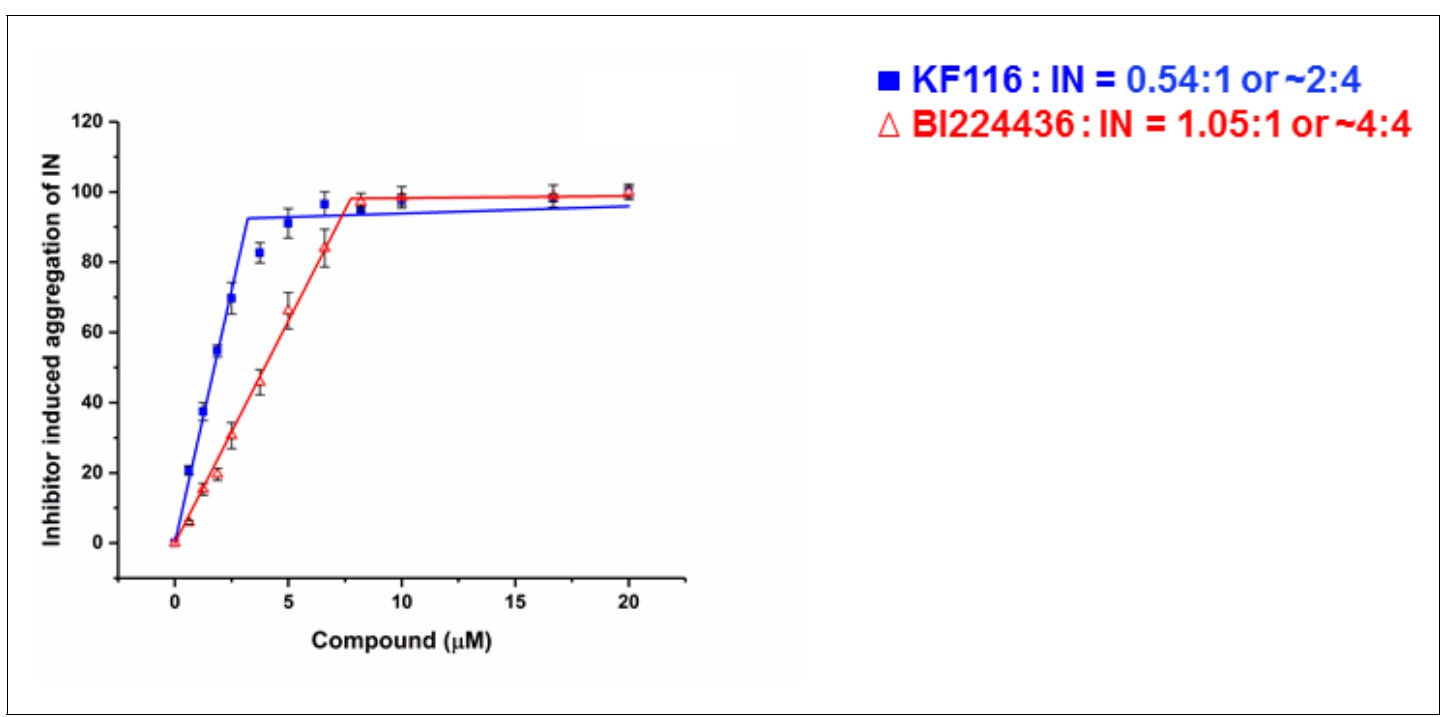

**Figure 2.** Stoichiometry of KF116 and BI224436 induced aggregation of IN. Quantitative analysis of ALLINI induced IN aggregation. The error bars indicate the standard deviation of three independent experiments. The stoichiometry for KF116:IN and BI224436:IN were determined using piecewise linear regression.

The online version of this article includes the following figure supplement(s) for figure 2:

**Figure supplement 1.** KF116 and BI224436 induced aggregation of IN.

ratio (*Figure 2* and *Figure 2—figure supplement 1*), similar to the prior crystal structure (*Gupta et al., 2016*).

## NTD and the α-helical linker connecting CCD with CTD are critically important for ALLINI induced higher-order INMultimerization

Previous studies from us and others (*Deng et al., 2016*; *Gupta et al., 2016*; *Shkriabai et al., 2014*) have elucidated the importance of CCD-ALLINI-CTD interactions but a role for the NTD in inhibitor induced higher-order multimerization has not been examined. The NTD was disordered and could not be resolved in the crystal structure of IN Y15A/F185H + GSK1264 (*Gupta et al., 2016*). Therefore, we compared how KF116 and BI224436 affected full-length WT IN and its various truncated constructs. DLS results in *Figure 3A and B*, show that KF116 and BI224436 promoted higher-order multimerization of only full-length IN, whereas all truncated protein constructs (CCD, NTD-CCD and CCD-CTD) exhibited marked resistance. In addition to DLS experiments, which monitored higher-order protein multimerization within the first 60 mins after the addition of ALLINIs to IN, we conducted aggregation assays that detected insoluble aggregates formed after ~16 hr incubation of IN with the inhibitors (*Figure 3C,D* and *Figure 3—figure supplement 1*). Addition of KF116 or BI224436 aggregated full-length WT IN in a dose dependent manner, whereas the CCD and NTD-CCD, both of which lacked the CTD, remained fully soluble even after incubation with 20 µM inhibitors (*Figure 3C,D* and *Figure 3—figure supplement 1*). These observations are consistent with the published results from our and other groups indicating that both the CCD and CTD are required for

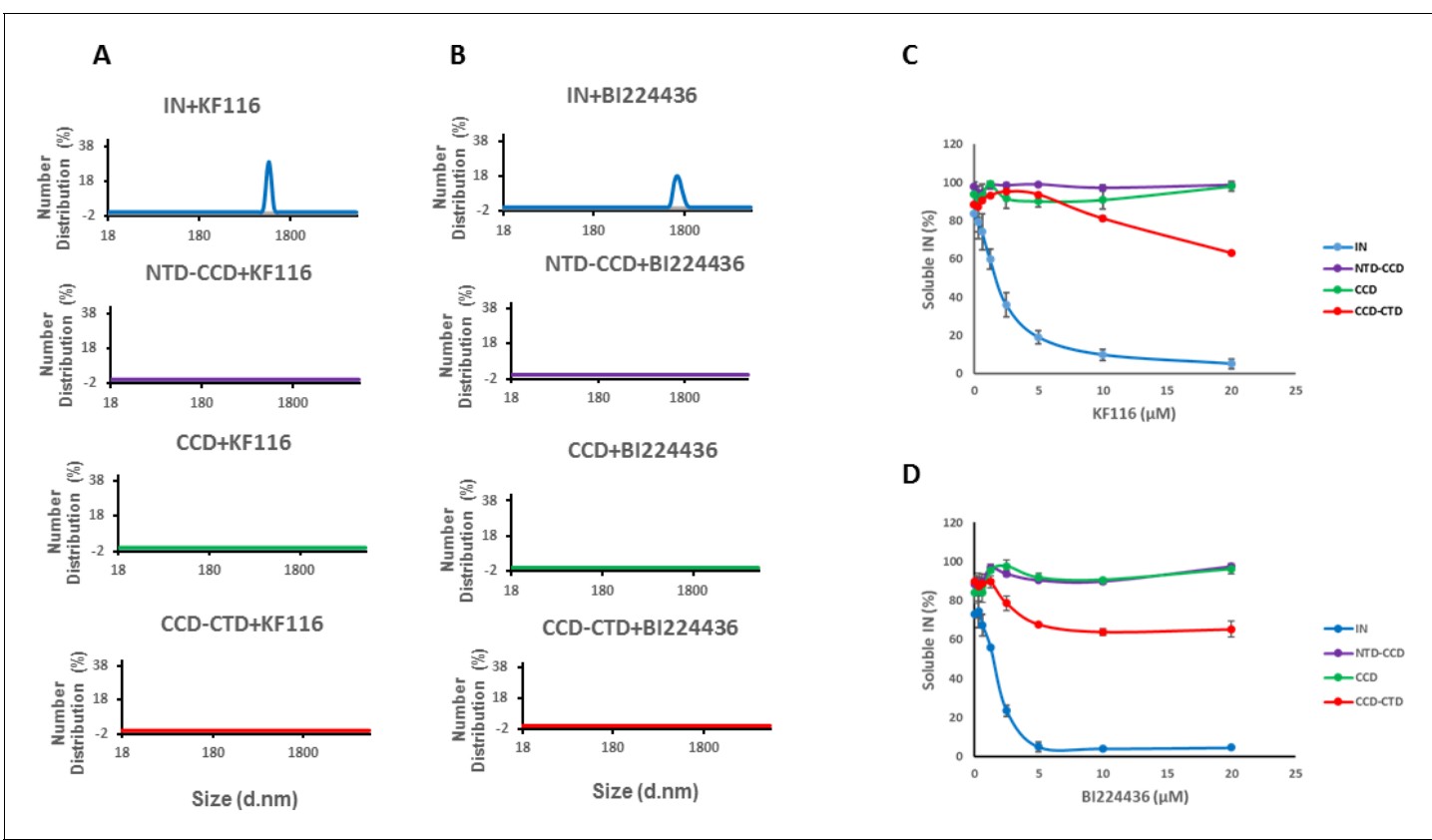

**Figure 3.** Roles of individual IN domains for ALLINI induced aggregation. (**A** and **B**) DLS analysis of 10 µM full length IN, NTD-CCD, CCD and CCD-CTD in the presence of 1 µM KF116 or BI224436. Representative results of two independent experiments at 30 mins time point are shown. DMSO control results for each respective experiment are shown in gray. (**C** and **D**) Quantitative analysis of KF116 and BI224436 induced aggregation of 5 µM full length IN, NTD-CCD, CCD and CCD-CTD by centrifugation-based aggregation assay. The error bars indicate the standard error of two independent experiments (see *Figure 3—figure supplement 1* for representative primary data).

The online version of this article includes the following figure supplement(s) for figure 3:

**Figure supplement 1.** Contributions of individual IN domains for ALLINI induced aggregation.

ALLINI induced higher-order IN oligomerization (*Deng et al., 2016*; *Gupta et al., 2014*; *Gupta et al., 2016*; *Shkriabai et al., 2014*). Interestingly, the CCD-CTD, which contained all inhibitor interacting interfaces but which lacked the NTD, also exhibited substantial resistance to both KF116 and BI224436 induced aggregation (*Figure 3C,D*). Of note, only residual aggregation of CCD-CTD was detected at higher KF116 and BI224436 concentrations (>10 µM), whereas these inhibitor concentrations fully precipitated full-length WT IN. Collectively, these findings (*Figure 3* and *Figure 3—figure supplement 1*) indicate that in addition to the CCD and CTD, the NTD is critically important for the inhibitor induced higher-order IN multimerization.

Unlike the CCD and CTD, the NTD does not directly interact with the inhibitors but this domain is essential for functional tetramerization of IN (*Hare et al., 2009*). Specifically, the NTD from one dimer engages with the CCD of another dimer to help form stable tetramers. In addition, the NTD interacts with the α-helical linker (200-222) connecting the CCD with the CTD. Therefore, to probe the significance of IN tetramerization for ALLINI induced higher-order IN multimerization, we targeted these protein regions (the NTD and the CCD-CTD linker) with site directed mutagenesis to specifically compromise functional IN tetramerization without affecting the direct sites of ALLINI binding. The H12A IN substitution, which destabilizes binding of the architecturally important $Zn^{2+}$, markedly altered IN oligomerization, yielding a mixture of monomers and higher-order oligomers (*Figure 4* and *Figure 4—figure supplement 1*). H12A IN was susceptible to BI224436 but exhibited marked resistance to KF116 (*Figure 4* and *Figure 4—figure supplement 2*). The K14A change destabilized NTD-CCD interactions needed for functional IN tetramerization and resulted in IN dimers (*Figure 4* and *Figure 4—figure supplement 1*). KF116 was completely inactive, whereas BI224436 remained active against K14A IN (*Figure 4* and *Figure 4—figure supplement 2*). Y15A IN, which yielded a mixture of dimers and monomers, was resistant to both KF116 and BI224436. Y15A/ F185H IN, which was utilized in the x-ray structure with quinoline-based compounds (*Gupta et al., 2016*) and which yielded dimers, was susceptible to quinoline-based BI224436 but this mutant was completely resistant to pyridine-based KF116 (*Figure 4* and *Figure 4—figure supplement 2*).

Next, we targeted the CCD-CTD α-helical linker by site directed mutagenesis (*Figure 4* and *Figure 4—figure supplements 1–4*). We incorporated a single Pro residue between T210 and K211 to introduce a kink in the α-helix connecting the CCD with the CTD. T210 +Pro IN was completely resistant to both KF116 and BI224436 indicating that the linear α-helix is critically important for accurate orientation of the CCD-inhibitor-CTD interfaces. Similarly, deletion of A205 within the α-helix rendered IN resistant to both KF116 and BI224436.

The substitutions in the NTD or the CCD-CTD connecting α-helix, both of which compromised functional tetramerization of IN, also impaired virus infectivity and inhibited IN catalytic activity in vitro (*Figure 4—figure supplements 3* and *4*) (*Hare et al., 2009*). Furthermore, these substitutions also adversely affected LEDGF/p75 binding with lesser effects seen for the T210 +Pro substitution, which is significantly distanced from LEDGF/p75 binding sites in the NTD or CCD (*Figure 4—figure supplement 3*). The control N222K substitution in the CCD-CTD connecting α-helix formed tetramers, retained WT levels of HIV-1 replication in cell culture, IN catalytic activity and LEDGF/p75 binding in vitro, and was fully susceptible to both KF116 and BI224436 induced higher-order IN multimerization (*Figure 4* and *Figure 4—figure supplements 1–4*).

Collectively, the experiments with select substitutions in the NTD and CCD-CTD linker, which allowed us to compromise functional IN tetramerization without directly affecting ALLINI binding sites, revealed a consistent correspondence between the effects of the substitutions on IN tetramer formation and the ability of KF116 to induce higher-order multimerization. Moreover, our findings further highlighted both key similarities and differences between pyridine- and quinoline-based compounds.

## Trans-complementation assays confirm KF116 selectivity for IN tetramers

Published structural studies (*Hare et al., 2009*) have revealed key interactions that govern IN tetramerization. For example, an ionic bond between E11 in the NTD of one dimer and K186 within the CCD of another dimer is important for functional tetramerization of IN (*Hare et al., 2009*) (*Figure 5A*). Individual E11K and K186E IN substitutions fully compromise protein tetramerization (without affecting IN dimerization) due to charge-charge repulsions. However, mixing two mutant proteins (E11K and K186E INs) in vitro allows for partial reconstitution of functional IN tetramers

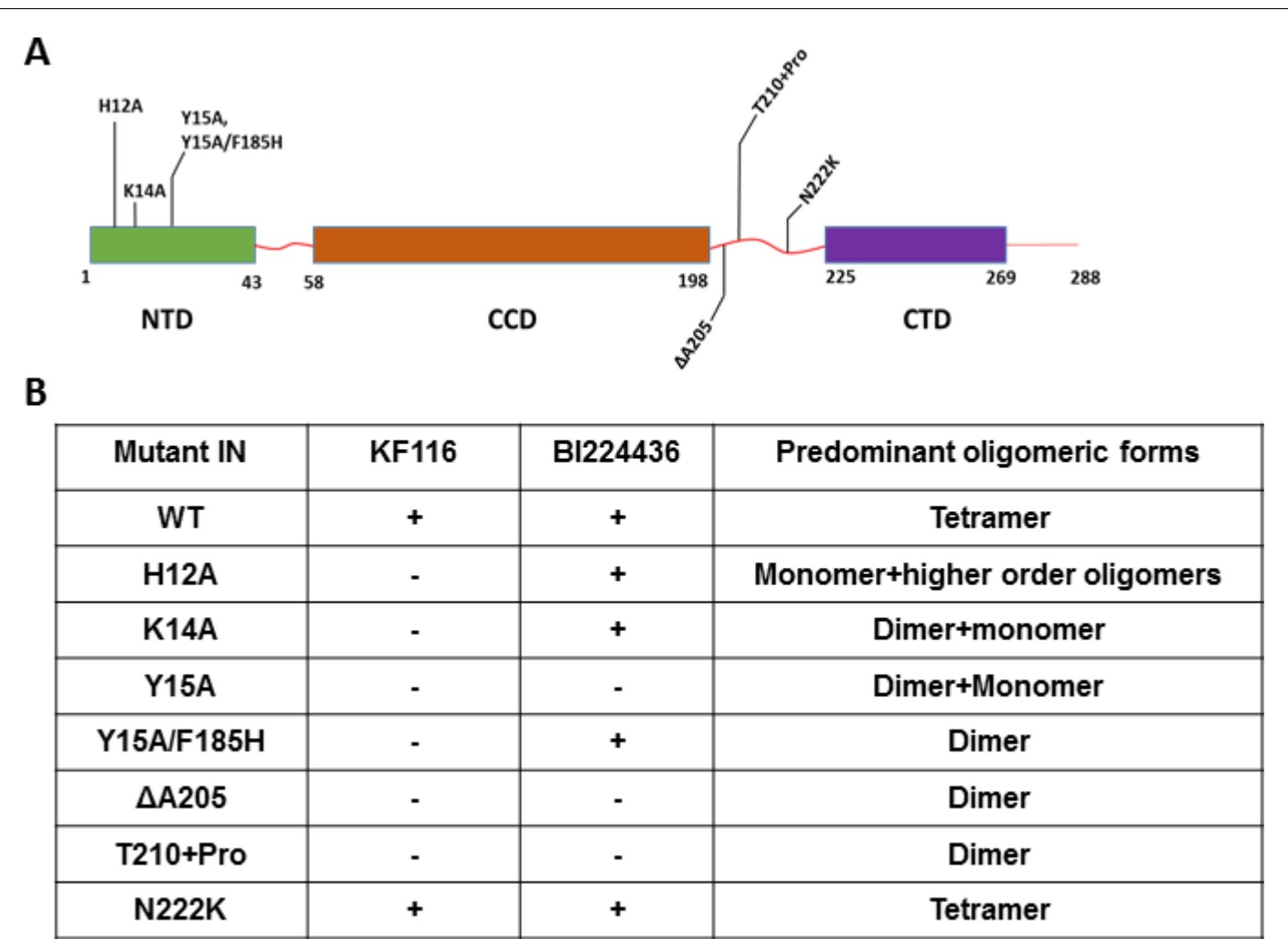

**Figure 4.** Probing the importance of the NTD and CCD-CTD linker for ALLINI induced higher-order multimerization of IN. (**A**) Schematic diagram of IN with indicated mutations in NTD and CCD-CTD linker regions. (**B**) Summary table of all IN mutants indicating their predominant multimeric form as analyzed by SEC and effects of ALLINI induced higher-order multimerization monitored by DLS. '+' and '- ' indicates susceptibility and resistance of the mutant proteins to ALLINI induced higher-order multimerization, respectively.

The online version of this article includes the following figure supplement(s) for figure 4:

**Figure supplement 1.** Multimeric forms of IN mutants.
**Figure supplement 2.** Higher-order IN multimerization induced by KF116 and BI224436.
**Figure supplement 3.** Biochemical characterization of IN mutants.
**Figure supplement 4.** Effect of IN substitutions on the infectivity of HIV-1$_{NL4-3}$.

through reversed charge-charge interactions (*Hare et al., 2009*). Indeed, the results in *Figure 5— figure supplement 1* show that individual E11K and K186E substitutions compromised the catalytic activities of IN and its ability to bind LEDGF/p75, whereas these functions of IN were substantially restored through reconstituting the two inactive proteins (E11K + K186E INs). Therefore, we used this powerful trans-complementation system for testing the significance of IN tetramerization for KF116 and BI224436 activity.

DLS results in *Figure 5B* revealed that addition of KF116 to either E11K or K186E IN dimers did not induce higher-order protein multimerization. However, when two resistant IN mutants (E11K and K186E) were mixed together to allow reconstitution of IN tetramers and then incubated with KF116 under identical reaction conditions, we observed formation of higher-order protein multimers

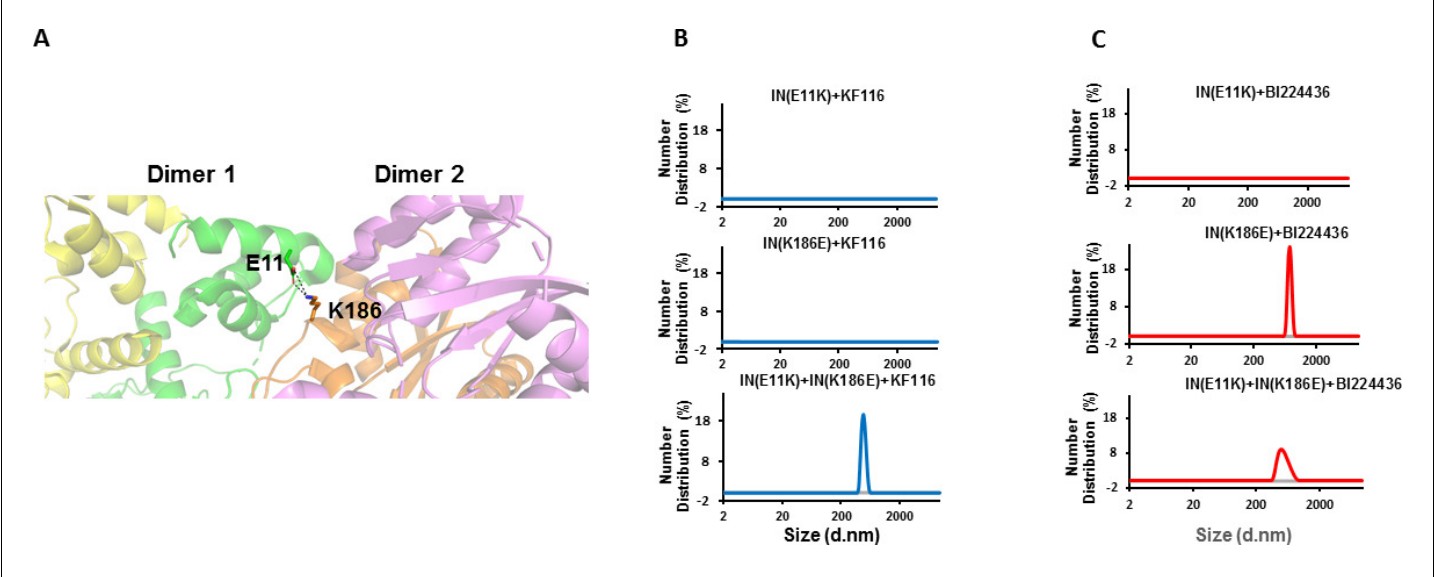

**Figure 5.** Trans-complementation of IN tetramer interface mutants to elucidate ALLINI preferences. (**A**) The salt bridge between E11 and K186 residues is highlighted in the context of IN tetramer interface. (**B**) DLS analysis of 200 nM E11K, K186E and E11K + K186E INs in the presence of 1 µM KF116 or BI224436 after 30 mins. DMSO controls with respective INs are shown in gray. Representative results of two independent experiments are shown. The online version of this article includes the following figure supplement(s) for figure 5:

**Figure supplement 1.** Trans-complementation of inactive mutants partially restores IN functions.

(*Figure 5B*). These findings provide another line of evidence that KF116 exhibits marked selectivity for IN tetramers.

In parallel experiments, BI224436 induced higher-order multimerization of dimeric K186E and reconstituted tetrameric E11K + K186E INs (*Figure 5C*), which provide further support for a broader specificity of BI224436 for tetramers and dimers. However, E11K IN was resistant to BI224436 suggesting that this substitution, in addition to destabilizing IN tetramers, could also potentially affect the proper orientation of the NTD required for BI224436 induced higher-order IN multimerization of the IN dimers. Accordingly, we also note that E11K was significantly more defective for IN catalytic activity than was K186E IN (*Figure 5—figure supplement 1*).

## Molecular modeling of ALLINI induced higher-order IN multimers

We used our experimental findings and available structural information to create molecular models for KF116 and BI224436 induced higher-order multimerization of WT IN. The full-length IN tetramer and dimer models were assembled based on the cryo-EM and x-ray crystal structures of lentiviral (HIV-1 and MVV) intasomes (tetramer model) and IN domains (dimer model) (*Ballandras-Colas et al., 2017*; *Chen et al., 2000*; *Gupta et al., 2016*; *Hare et al., 2009*; *Passos et al., 2017*; *Wang et al., 2001*). Next, the HADDOCK program, a data-driven protein-protein docking algorithm (*Dominguez et al., 2003*; *van Zundert et al., 2016*), was used to create molecular models for KF116 mediated tetramer-tetramer interactions (*Figure 6A*). For this the crystal structure of KF116 bound to HIV-1 IN CCD dimer was used to establish the head-to-tail interactions between KF116 bound to the CCD-CCD dimer of one tetramer and CTD of another tetramer (*Sharma et al., 2014*). Docking simulations using HADDOCK generated top ranked symmetric head-to-tail architecture, with the two tetramers connected by two CCD-KF116-CTD interfaces (*Figure 6A,B*). This model is fully consistent with the experimentally observed stoichiometry of 2:4 for KF116:IN. The KF116 mediated buried surface area (BSA) between two tetramers was ~2000 Å$^2$–3200 Å$^2$. Initial HADDOCK simulations yielded the BSA of ~2000 Å$^2$ and further optimization of the symmetric interactions resulted in an improved BSA of 3200 Å$^2$. The optimization was performed by specifying all interface residues from the initial HADDOCK docked structure as the input for the second HADDOCK run. Strikingly, when we used IN dimers instead of tetramers as building blocks, the

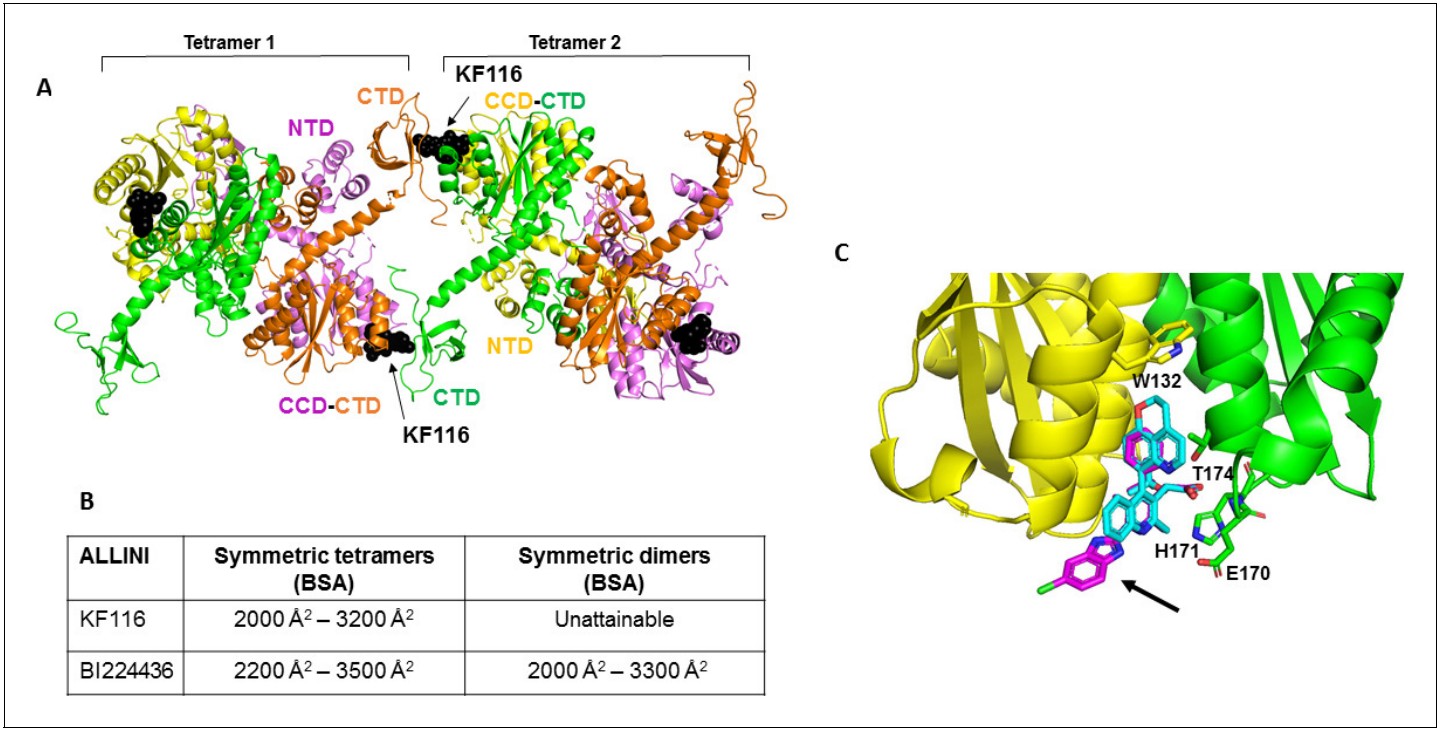

**Figure 6.** Structural analysis of KF116 and BI224436 interactions with IN. (**A**) The top ranked model for symmetric tetramer-KF116-tetramer interactions. Each protomer is distinctly colored (green, yellow, violet, orange). Each domain is assigned to its respective protomer, as previously proposed for Maedi Visna IN (PDB: 3HPH)(*Hare et al., 2009*). (**B**) Buried surface areas (BSA) of KF116 and BI224436 induced higher-order IN multimers; (**C**) overlay of the crystal structures of KF116 (PDB: 4O55) and BI224436 bound to IN CCD$_{F185H}$. KF116 and BI224436 are shown in magenta and cyan respectively. The arrow points to the benzimidazole group in KF116.

The online version of this article includes the following figure supplement(s) for figure 6:

**Figure supplement 1.** Molecular modeling of BI224436 induced higher-order IN multimerization.

HADDOCK program failed to yield symmetric dimer:KF116:dimer assemblies. These findings are consistent with experimental results demonstrating marked selectivity of KF116 for IN tetramers.

For comparison, we have also built multimer models containing BI224436 (*Figure 6—figure supplement 1*). For this, we have first solved the x-ray structure of BI224436 bound to IN CCD (*Figure 6C* and *Supplementary file 1* (Table S1)). In common with all other potent ALLINIs, the carboxylic acid moiety of BI224436 maintains important hydrogen bonding interactions with the backbone amides of E170 and H171. In addition, the hydrogen bonding interaction of the ether oxygen of the *tert*-butoxy and the side chain of T174 is conserved. However, there are the following important differences between BI224436 and KF116 binding (*Figure 6C*): i) the bulky nature of the tricyclic ring of BI224436 fully fills the hydrophobic pocket of the CCD dimer and extends closer to W132, which caps the pocket. For comparison, KF116 contains a much smaller benzyl substitution at this position; ii) the benzimidazole group, which is both unique and essential for KF116 activity, projects outside of the v-shaped pocket, whereas BI224436 is more deeply positioned within the pocket.

Next, we used our x-ray structure (*Figure 6C*) to build models of BI224436 interactions with full-length IN (*Figure 6—figure supplement 1*). Symmetric multimers can be generated regardless of whether IN tetramers or dimers are used as building blocks suggesting that unlike KF116, BI224436 can recruit *both* tetramers and dimers for higher-order IN multimerization. These in silico findings are fully consistent with the experimental results indicating that unlike KF116, which is highly selective for IN tetramers, BI224436 exhibits a broader specificity for tetramers and dimers (*Figure 1* and *Figure 1—figure supplement 1*).

Our molecular models (*Figure 6A* and *Figure 6—figure supplement 1*) are also consistent with experimental data showing the importance of the NTD for inhibitor induced higher-order IN oligomerization. Specifically, in the symmetric tetramer-KF116-tetramer model (*Figure 6A*) while the NTD

does not directly engage the inhibitor, this domain plays two key architectural roles. First, the NTD of one dimer interacts with the CCD of another dimer to stabilize IN tetramers (*Hare et al., 2009*). Second, the NTD interacts with the linear α-helix (200-222) connecting the CCD with CTD, which in turn could affect correct orientation of the CTD for inhibitor induced head-to-tail interactions. This latter interaction of the NTD with the CCD-CTD linker is also seen in the context of symmetric tetramer-BI224436-tetramer and dimer-BI224436-dimer assemblies (*Figure 6—figure supplement 1*). Thus, these modeling results are fully consistent with our experimental results indicating that NTD could contribute to both KF116 and BI224436 induced higher-order IN multimerization.

### The (-)-KF116 enantiomer exhibits high potency and metabolic stability

Previously, we have reported antiviral activity of ~24 nM for racemic KF116 in single replication cycle assays (*Sharma et al., 2014*). We have now synthesized (-) and (+)-KF116 enantiomers and assayed their antiviral activities during multiple rounds of HIV-1 replication in MT-4 cells. (-)-KF116 exhibited an $IC_{50}$ of ~7 nM, which was ~30 times more potent than its (+) counterpart (*Figure 7A* and *Figure 7—figure supplement 1A*).

Next, we evaluated the metabolic stability of (-)-KF116 using rat and human liver microsomes (*Figure 7—figure supplement 1B*). We probed in vitro Cytochrome (CYP) P450 activity in the presence of co-factor NADPH (*Wempe and Anderson, 2011*; *Wempe et al., 2012a*; *Wempe et al., 2012b*) and monitored ALLINI stability by LC-MS. In vitro half-life measurements and calculated intrinsic clearance values in *Figure 7—figure supplement 1B* show that control compounds Verapamil, Domperidone and Chlorpromazine were metabolized as expected while ALLINIs displayed

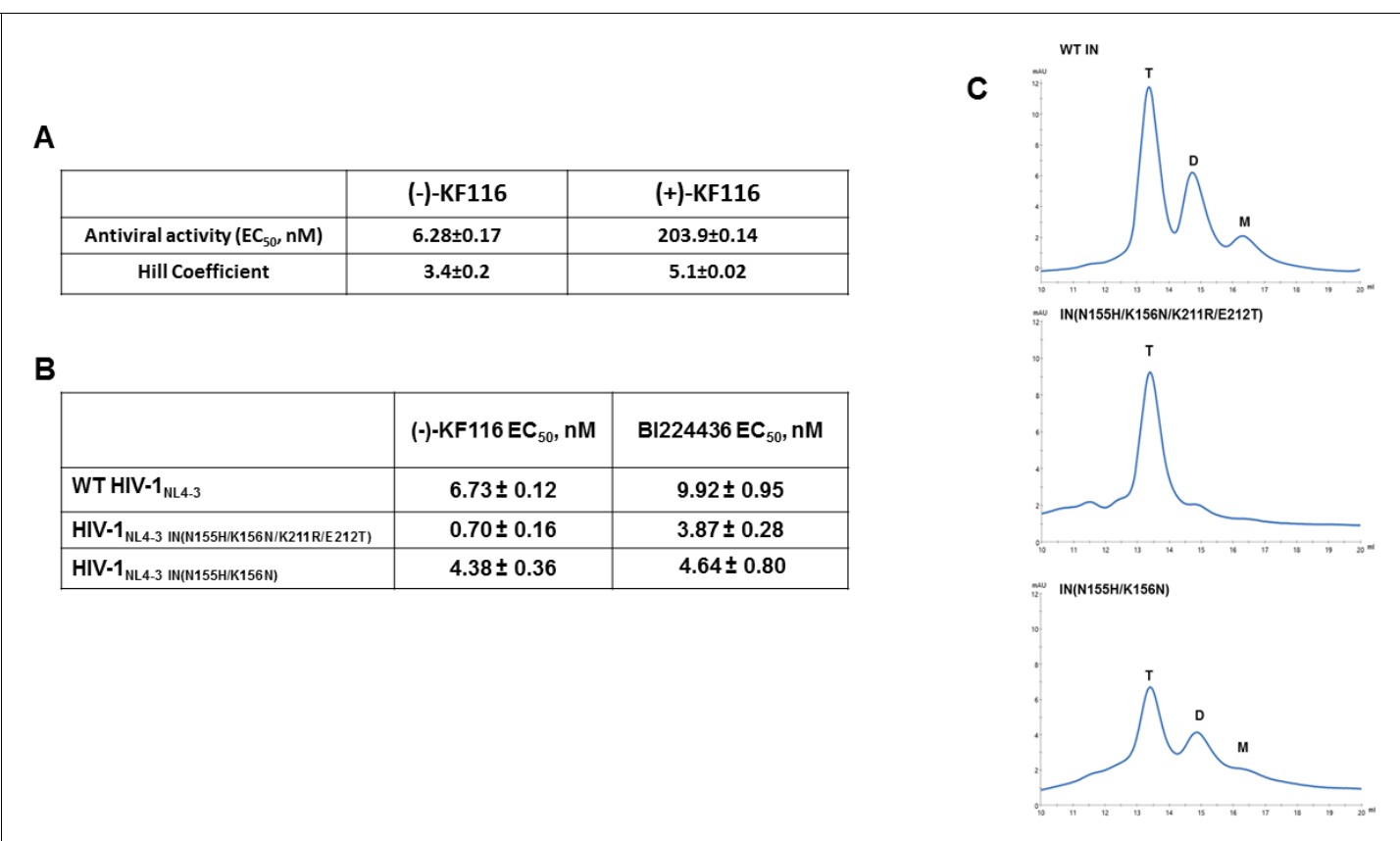

**Figure 7.** Antiviral activities of ALLINIs. (**A**) Antiviral activities of (-) and (+)- KF116 against WT virus. (**B**) Antiviral activities of KF116 and BI224436 against DTG resistant quadruple and double mutant viruses. The error is the S.D. of three independent experiments. (**C**) SEC analysis of mutant INs.
The online version of this article includes the following figure supplement(s) for figure 7:

**Figure supplement 1.** Comparative analysis of (+) and (-) enantiomers of KF116.

excellent metabolic stability toward CYP oxidation with (-)-KF116 exhibiting superior properties compared with racemic KF116 and quinoline-based BI224436.

## (-)-KF116 exhibits enhanced activities against a clinically relevant DTG resistant virus

Second generation INSTIs such as DTG, which bind at the IN catalytic site in the presence of viral DNA, display a high genetic barrier to resistance. Therefore, the drug resistance phenotypes emerging in the clinic in response to second generation INSTIs reveal complex resistance profiles with IN substitutions often seen outside of the inhibitor binding site. For example, a recent clinical study revealed that failure of DTG treatment in patients was observed with concomitant appearance of IN N155H/K211R/E212T substitutions on the background of the K156N polymorphic mutation (*Malet et al., 2018*). N155 and K156 are within the CCD, in close proximity to the IN active site. In contrast, K211 and E212 are significantly distanced from the DTG binding site and instead these residues are located in the CCD-CTD connecting α-helix implicated by our modeling and site directed mutagenesis studies as critically important for KF116 induced higher-order IN multimerization (*Figures 4* and *6*). Therefore, we wanted to examine (-)-KF116 activity with respect to the DTG resistant virus. Interestingly, KF116 displayed sub-nM activity and was ~10 fold more active against HIV-1 containing IN N155H/K156N/K211R/E212T substitutions compared with the WT virus (*Figure 7B*). In control assays, KF116 was similarly active against WT HIV-1 and the mutant virus containing only the CCD (N155H/K156N) substitutions (*Figure 7B*).

Based on our biochemical assays highlighting the significance of the CCD-CTD connecting α-helix for functional tetramerization of IN and KF116 activities, we wanted to also test how K211R/E212T substitutions affected oligomerization of recombinant IN. Our SEC data (*Figure 7C*) show that recombinant IN(N155H/K156N/K211R/E212T) is almost exclusively tetramer, whereas IN(N155H/K156N), which is similar to WT IN (*Figure 7C*), forms a mixture of tetramers, dimers and monomers. These findings provide further evidence for a strong correlation between IN tetramers and KF116 activity (*Figure 7B and C*) in the context of therapeutically important DTG resistant mutant viruses.

## ALLINI treatment compromises the association of IN with the viral core

ALLINI treatment induces higher-order multimerization of IN in virions, which compromises the ability of IN to bind the viral RNA genome and causes mislocalization of RNPs outside of the conical core made of capsid proteins (CA) (*Jurado and Engelman, 2013*; *Kessl et al., 2016*). However, it is not clear whether mislocalized IN complexes in ALLINI-treated samples remain associated with the viral CA core and whether these complexes traffic together to the nucleus. To address this question, we employed live-cell imaging to visualize IN co-trafficking with the viral core. HIV-1 pseudoviruses were co-labeled with IN-mNeonGreen (INmNG) and cyclophilinA (CypA) fused to DsRed (CypA-DsRed), a marker of the viral CA protein (*Francis et al., 2016*; *Francis and Melikyan, 2018*). This co-labeling strategy enables real-time tracking of viral complexes in living cells and visualization of distinct steps of HIV-1 entry, uncoating, nuclear import and integration (*Francis et al., 2016*; *Francis and Melikyan, 2018*). We have previously shown that productive uncoating (loss of CypA-DsRed/CA) that leads to infection occurs at the nuclear pore, prior to delivery of the IN complexes into the nucleus. By contrast, premature loss of CypA-DsRed/CA in the cytoplasm exposes the remaining IN complexes and leads to their proteasomal degradation (*Francis and Melikyan, 2018*).

Fluorescent viruses were produced in the presence of ALLINIs, or, in control experiments, in the presence of DMSO or the INSTI RAL. The fate of single INmNG- and CypA-DsRed-labeled complexes during virus entry was examined in HeLa-derived cells depleted of CypA (PPIA-/-) by live-cell imaging. Cells depleted of CypA were chosen to reduce premature uncoating of the incoming postfusion CA cores (*Francis and Melikyan, 2018*) and thereby extend the window of observation for assessing the fate of INmNG in living cells. The viral particles produced in the presence of (+/-)-KF116, (-)-KF116 and BI224436 were compared with DMSO and RAL controls (*Figure 8* and *Figure 8—figure supplement 1*). Imaging of single viral complexes bound to a cover glass before or after 5 min of virus entry into target cells showed similar extents of colocalization between CypA-DsRed and INmNG puncta in all samples (*Figure 8—figure supplement 1*). However, imaging of these virions in the same target cells at 90 min post-infection revealed a marked drop in the number of co-labeled particles (primarily due to a loss of IN signal) for viruses produced in the presence of

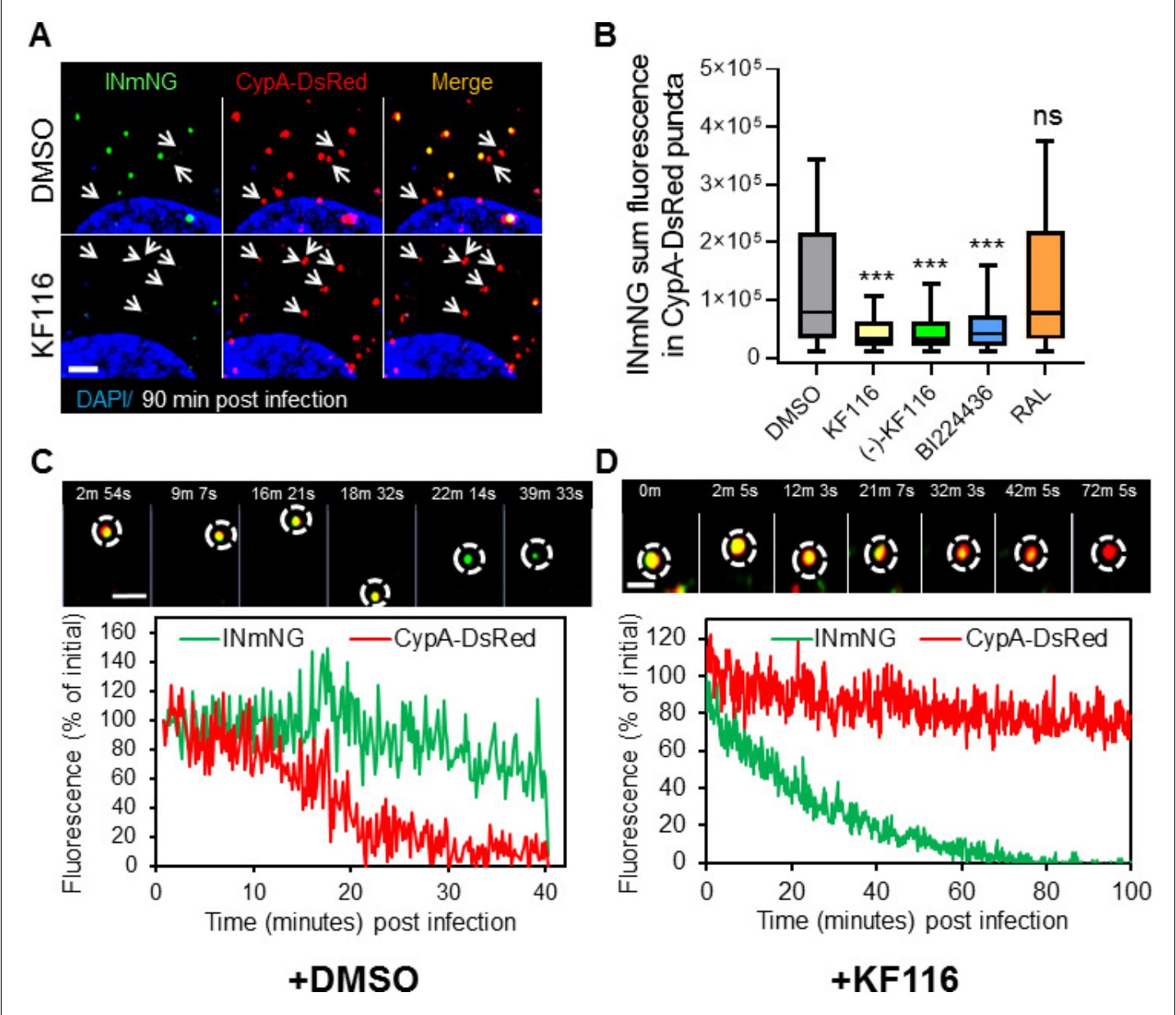

**Figure 8.** VSV-G pseudotyped fluorescent HIV-1 labeled with Vpr-IN-mNeonGreen (INmNG,a marker of viral complex) and CypA-DsRed (CA marker). Viruses produced in the presence of 5 μM ALLINIs or raltegravir (RAL) or left untreated (DMSO) control were used to infect TZM-bl PPIA-/- cells (MOI 0.08) for 90 min. Cells were imaged on the Zeiss LSM880 using Fast-Optimal AiryScan settings, which enables sensitive detection and tracking of viral complexes with high temporal resolution and minimal photobleaching. (A) Images of CypA-DsRed labeled cores retaining INmNG signal for DMSO treated samples and loss of INmNG signal from the viral CA cores, leaving CypA-DsRed puncta for KF116 treated samples at 90 min post infection. Arrows point to single labeled CypA-DsRed complexes. (B) The sum fluorescence of INmNG within CypA-DsRed/CA puncta is plotted. (C–D) Time-lapse imaging of INmNG/CypA-DsRed labeled virus. Single particles losing the CypA-DsRed or INmNG signal were manually annotated and tracked. (C) Images and fluorescent intensity traces showing uncoating (loss of CypA-DsRed prior to INmNG loss) for single viral particle produced in the presence of DMSO. (D) Images and fluorescent intensity traces showing a gradual INmNG signal loss from a CypA-DsRed/CA puncta for viruses produced in the presence of KF116 (5 μM). Scale bar in (A, C and D) =2 μm. Statistical significance in (B) was determined by comparing respective samples with the DMSO control using a pair-wise student t-test, ***=p < 0.0001. p values > 0.5 were considered not significant (ns).

The online version of this article includes the following figure supplement(s) for figure 8:

**Figure supplement 1.** VSV-G pseudotyped HIV-1 viral particles fluorescently labeled with INmNG and CypA-DsRed in the presence of indicated treatments.

**Figure supplement 2.** Schematic summary of the experimental results.

ALLINIs, but not for control viruses prepared in the presence of DMSO or RAL (*Figure 8A–B* and *Figure 8—figure supplement 1*).

We next analyzed the pattern of INmNG loss from CypA-DsRed/CA labeled cores upon entry of ALLINI pretreated HIV-1 pseudoviruses. Single particle tracking revealed that DMSO-treated control viruses undergo uncoating manifested in a loss of CypA-DsRed from INmNG labeled complexes (*Figure 8C*). In agreement with our previous studies showing proteasomal degradation of post-uncoating IN complexes in the cytoplasm (*Francis et al., 2016*; *Francis and Melikyan, 2018*), we observed a gradual decay of the INmNG signal *after* the loss of CypA-DsRed signal from the viral complexes produced in the presence of the control DMSO solvent (*Figure 8C*). In contrast, ALLINI treated virions lost the INmNG signal *prior* to the loss of CypA-DsRed/CA (*Figure 8D*). These findings suggest that ALLINI induced INmNG aggregates are readily accessible to proteasomal degradation in the cytoplasm of target cells due likely to their mislocalization outside of the protective CA cores (*Figure 8—figure supplement 2*), which in turn precludes productive infection.

## Discussion

We present several lines of evidence demonstrating that IN tetramers are the authentic targets for the lead pyridine-based KF116: (i) the inhibitor selectively induced higher-order multimerization of WT IN tetramers but not its lower oligomeric states; (ii) the NTD and α-helical CCD-CTD linker (residues 200–222), which do not directly interact with ALLINIs but which are essential for functional tetramerization of full length IN, are also critically important for KF116 induced higher-order IN multimerization; (iii) single amino acid substitutions that compromise IN tetramerization also adversely affect KF116 activity; (iv) trans-complementation of two KF116 resistant IN dimers allows a partial reconstitution of IN tetramers, which in turn become susceptible to KF116 induced higher order multimerization; (iv) a positive correlation between increased antiviral activity of KF116 against the clinically relevant DTG resistant mutant virus and enhanced propensity of the corresponding mutated IN to form tetramers.

While the overall mechanism of action of pyridine- and quinoline-based ALLINIs are very similar, our findings highlight important differences between KF116 and BI224436. The most notable is that BI224436 exhibits a broader specificity for both IN tetramers and dimers unlike KF116, which is highly selective for IN tetramers. This correlates well with a prior NMR study that indicated that pyridine based ALLINIs exhibited higher binding affinity for IN tetramers compared to dimers (*Fader et al., 2014a*). Furthermore, BI224436, unlike KF116, is active against IN Y15A/F185H dimers. This was likely a key factor that enabled the generation of a successful x-ray structure for a quinoline-based GSK1264 (which is very similar to BI224436) bound to IN Y15A/F185H (*Gupta et al., 2016*). This structure provided breakthrough information about ALLINI induced head-to-tail interactions, where GSK1264 is seen sandwiched between a CCD dimer and the CTD of another dimer.

Initial structural clues about differential interactions of pyridine- vs quinoline-based inhibitors with various oligomeric forms of WT IN come from our crystal structures of these compounds bound to the CCD dimer (*Figure 6C*). Superimposition of IN CCD co-crystal structures with KF116 or BI224436 revealed both overlapping interactions as well as distinct contacts for each inhibitor with the CCD (*Figure 6C*). BI224436 (similarly to its very close analog GSK1264) is positioned deeper inside the v-shaped pocket (*Figure 6C*), which could allow symmetrical CCD-BI224436-CTD interactions between both IN dimers and tetramers. In contrast, the bulky benzimidazole group, which is both unique and essential for KF116 activity, extends from the v-shaped pocket (*Figure 6C*) and could force the incoming CTD to accept a slightly different orientation compared to BI224436 or GSK1264. This, in turn may hamper the dimer-dimer symmetry seen with GSK1264 (*Gupta et al., 2016*) but instead optimally position two IN tetramers for symmetrical CCD-KF116-CTD interactions (*Figure 6A*).

We show that in addition to direct CCD-ALLINI-CTD interactions, the NTD and α-helical CCD-CTD linker (200-222) are critically important for ALLINI induced higher-order multimerization. These protein segments do not directly interact with KF116 or BI224436 but are essential for functional tetramerization of IN. Within each IN tetramer, the NTD interacts with the CCD-CTD α-helical linker as well as the CCD of another dimer unit. These dual interactions allow the NTD to effectively stabilize the conformation of the CCD-CTD α-helical linker as well as the entire tetramer against thermal fluctuations. In turn, such conformational rigidity could facilitate symmetric tetramer-ALLINI-tetramer

assemblies. This notion is supported by our mutagenesis experiments (*Figure 4*) demonstrating that substitutions in the NTD and α-helical CCD-CTD linker (200-222) that compromise assembly of stable tetramers also adversely affect KF116 induced higher-order oligomerization.

Since both KF116 and BI224436 are highly potent during virion maturation and only IN tetramers can be targeted by both compounds, we hypothesize that the predominant form of IN in virions resides as a tetramer. It should also be noted that IN tetramers bind viral RNA and are the essential building blocks for super molecular assembly of lentiviral intasomes (*Ballandras-Colas et al., 2017*; *Kessl et al., 2016*; *Passos et al., 2017*). These functionally critical IN tetramers are selectively and preferentially targeted by KF116 and BI224436, respectively (*Figure 1* and *Figure 1—figure supplement 1*).

ALLINI induced higher-order multimerization of IN in virions inhibits IN interactions with the viral RNA and results in mislocalization of RNPs outside of the translucent CA cores (*Fontana et al., 2015*; *Jurado et al., 2013*; *Kessl et al., 2016*). While these eccentric virions can effectively enter the target cells, RNPs and IN are prematurely degraded during the early phase of infection (*Madison et al., 2017*). To gain further insight into the mechanism of action of ALLINIs we have used INmNG and CypA-DsRed labeling of CA and monitored co-trafficking of these viral proteins during the early phase of infection. As expected (*Francis et al., 2016*; *Francis and Melikyan, 2018*), in the absence of inhibitors, we observed the initial decay of CypA-DsRed signal likely due to the uncoating of CA cores followed by subsequent, substantially slower decay of INmNG signal. In sharp contrast, the ALLINI treated viruses exhibited a remarkable loss of INmNG signal from single particle post-fusion viral cores that were yet to complete uncoating. The schematic summary of these experimental observations are depicted in *Figure 8—figure supplement 2A*. In turn, the pre-mature loss of IN from the viral CA cores during infection of target cells is likely to be a consequence of aberrant packaging of ALLINI induced higher-order IN multimers within virions (*Figure 8—figure supplement 2B*). Collectively, our current findings together with published results (*Kessl et al., 2016*) suggest that binding of functional IN tetramers to the viral RNA genome are important for effective packaging of these viral components within conical CA cores and formation of the infectious virions (*Figure 8—figure supplement 2*). In contrast, ALLINI treatment results in aggregation and separation of IN from RNPs. As a result, both IN-ALLINI aggregates and RNPs lacking IN are mislocalized outside of the protective CA cores and yield eccentric, non-infectious virions (*Figure 8—figure supplement 2B*).

The present studies argue for the further development of KF116 for its potential translation into clinical applications. Arguably, the most exciting finding is that KF116 exhibits enhanced, sub-nM activity against therapeutically important DTG resistant HIV-1$_{NL4-3(IN\ N155H/K156N/K211R/E212T)}$. These substitutions have recently been identified in a patient receiving DTG monotherapy and follow up cell culture based virology experiments confirmed substantial resistance of HIV-1$_{NL4-3(IN\ N155H/K156N/K211R/E212T)}$ to DTG (*Malet et al., 2018*). Our interest in this particular resistant phenotype has been prompted by the presence of the two K211R/E212T substitutions in the α-helical CCD-CTD linker. Differing from artificial mutations (ΔA205 and T210 + Pro) introduced in this segment, which both impaired proper IN tetramerization and conferred marked resistance to KF116 (*Figure 4*), the K211R/E212T substitutions instead stabilized IN tetramers and resulted in enhanced antiviral activity of KF116 (*Figure 7*). These results argue strongly for the positive correlation between functional IN tetramerization and the ability of KF116 to induce higher-order IN oligomerization. It should also be noted that a previous study has shown that combinations of ALLINIs and INSTIs exhibited additive or even synergetic effects to inhibit HIV-1 in cell culture (*Christ et al., 2012*). Our observation that (-)-KF116 exhibits substantially enhanced, sub-nM activity with respect to DTG resistant HIV-1$_{NL4-3(IN\ N155H/K156N/K211R/E212T)}$ provides further support for potential clinical benefits of combining these two inhibitors to treat HIV-1 infected patients. New therapies that target the novel function of HIV-1 IN are critical for successful treatment of HIV-1 in patients and our work will help to inform future studies as ALLINIs translate from the lab to the clinic.

## Materials and methods

**Key resources table**

*Continued on next page*

*Continued*

| Reagent type (species) or resource | Designation | Source or reference | Identifiers | Additional information |
|---|---|---|---|---|
| Reagent type (species) or resource | Designation | Source or reference | Identifiers | Additional information |
| Strain, strain background (*Retroviridae*) | HIV-1$_{NL4-3}$ | Other | | Replication competent HIV-1 virus particles produced from pNL4-3 plasmid in HEK293T producing cells; laboratory adapted HIV-1 strain |
| Strain, strain background (*Retroviridae*) | HIV-1$_{NL-lucE-R+VSVg}$ | Other | | Replication incompetent HIV-1 virus particles produced from pNL-luc. E-R+and pMD.G plasmids in HEK293T producing cells; laboratory adapted HIV-1 strain pseudotyped with VSV glycoprotein |
| Genetic reagent (*Retroviridae*) | HIV-1$_{NL4-3}$ IN (K14A) | Other | | Replication competent HIV-1 virus particles containing the mutation K14A in the Integrase ORF |
| Genetic reagent (*Retroviridae*) | HIV-1$_{NL4-3}$ IN (Y15A) | Other | | Replication competent HIV-1 virus particles containing the mutation Y15A in the Integrase ORF |
| Genetic reagent (*Retroviridae*) | HIV-1$_{NL4-3}$ IN (T210 + Pro) | Other | | Replication competent HIV-1 virus particles containing the mutation T210 + Pro in the Integrase ORF |
| Genetic reagent (*Retroviridae*) | HIV-1$_{NL4-3}$ IN (N222K) | Other | | Replication competent HIV-1 virus particles containing the mutation N222K in the Integrase ORF |
| Genetic reagent (*Retroviridae*) | HIV-1$_{NL-lucE-R+ N155H/K156NVSVg}$ | Other | | Replication incompetent, VSVg pseudotyped HIV-1 virus particles containing the following mutations in the IN ORF: N155H/K156N |
| Genetic reagent (*Retroviridae*) | HIV-1$_{NL-lucE-R+ N155H/K156N/K211R/E212TVSVg}$ | Other | | Replication incompetent, VSVg pseudotyped HIV-1 virus particles containing the following mutations in the IN ORF: N155H/K156N/K211R/E212T |
| Cell line (*H.sapiens*) | HeLa | ATCC | ATCC CCL-2 | |
| Cell line (*H.sapiens*) | Hek293T | ATCC | ATCC CRL-3216 | |
| Cell line (*H.sapiens*) | TZM-bl | NIH AIDS Reagent Program | NIH AIDS Reagent Program: 8129 | The reagent was obtained through the NIH AIDS Reagent Program, Division of AIDS, NIAID, NIH: TZM-bl cells (Cat# 8129) from Dr. John C Kappes, and Dr. Xiaoyun Wu |

*Continued on next page*

*Continued*

| Reagent type (species) or resource | Designation | Source or reference | Identifiers | Additional information |
|---|---|---|---|---|
| Cell line (H.sapiens) | TZM-bl PPIA(-/-) | *Francis and Melikyan, 2018* | | |
| Cell line (H.sapiens) | MT-4 | NIH AIDS Reagent Program | NIH AIDS Reagent Program: 120 | The reagent was obtained through the NIH AIDS Reagent Program, Division of AIDS, NIAID, NIH: MT-4 from Dr. Douglas Richman |
| Biological sample (Rat) | Sprague-Dawley rat liver microsomes | Xenotech LLC | Xenotech LLC: 1510115 | Pool of 500, male; 20 mg/mL |
| Biological sample (H.sapiens) | Human Liver Microsomes | Xenotech LLC | Xenotech LLC: 1610016 | Pool of 50, mixed gender; 20 mg/mL |
| Recombinant DNA reagent | pNL4-3 (plasmid) | *Adachi et al., 1986* | | |
| Recombinant DNA reagent | pNL4-3 IN (K14A) (plasmid) | This paper | | Site-directed mutagenesis in pNL4-3 to generate IN K14A mutant |
| Recombinant DNA reagent | pNL4-3 IN (Y15A) (plasmid) | This paper | | Site-directed mutagenesis in pNL4-3 to generate IN Y15A mutant |
| Recombinant DNA reagent | pNL4-3 IN (T210 + Pro) (plasmid) | This paper | | Site-directed mutagenesis in pNL4-3 to generate IN T210 + Pro mutant |
| Recombinant DNA reagent | pNL4-3 IN (T222K) (plasmid) | This paper | | Site-directed mutagenesis in pNL4-3 to generate IN T222K mutant |
| Recombinant DNA reagent | pNL-luc.E-R+ (plasmid) | *Connor et al., 1995* | | |
| Recombinant DNA reagent | pNL-luc.E-R+N 155 H/K156N (plasmid) | This paper | | Site-directed mutagenesis in pNL-luc.E-R+to generate IN N155H/K156N mutant |
| Recombinant DNA reagent | pNL-luc.E-R+N155 H/K156N/K211R/E212T (plasmid) | This paper | | Site-directed mutagenesis in pNL-luc.E-R+to generate IN N155H/K156N/K211R/E212T mutant |
| Recombinant DNA reagent | pMD.G (plasmid) | *Naldini et al., 1996* | | |
| Recombinant DNA reagent | Vpr-INmNeon Green (INmNG) | *Francis and Melikyan, 2018* | | |
| Recombinant DNA reagent | CypA-DsRed | *Francis et al., 2016; Francis and Melikyan, 2018* | | |
| Recombinant DNA reagent | pHIVeGFP-deltaEnv | *Francis and Melikyan, 2018* | | |
| Recombinant DNA reagent | Recombinant IN (pET-15b) | *Larue et al., 2012* | | All the mutations were carried out by site directed mutagenesis in IN coding region of pET-15b |
| Recombinant DNA reagent | Recombinant IN domains: NTD-CCD, CCD and CCD-CTD (pET-15b) | This paper | | Wild type NL4-3 IN domains were constructed by site directed mutagenesis from pET-15b truncated IN domains containing solubilizing mutants (*Larue et al., 2012*) |
| Recombinant DNA reagent | Recombinant LEDGF (pLEDGF) | *Tsiang et al., 2009* | | |

*Continued on next page*

*Continued*

| Reagent type (species) or resource | Designation | Source or reference | Identifiers | Additional information |
|---|---|---|---|---|
| Commercial assay or kit | p24 ELISA | Zeptometrix | Zeptometrix: 0801111 | |
| Commercial assay or kit | MycoscopeMycoplasm PCR detection kit | Genlantis | Genlantis: MY01100 | |
| Commercial assay or kit | CellTiter-Glo | Promega Biosciences Inc | Promega Biosciences Inc: G7571 | |
| Commercial assay or kit | Luciferase Assay System | Promega Biosciences Inc | Promega Biosciences Inc: E1500 | |
| Commercial assay or kit | QuikChange XL site directed mutagenesis kit | Agilent | Agilent: 200516 | |
| Commercial assay or kit | QuikChange XL II site directed mutagenesis kit | Agilent | Agilent: 200522 | |
| Commercial assay or kit | LANCE Europium-streptavidin for HTRF assay | PerkinElmer | PerkinElmer: AD0062 | |
| Commercial assay or kit | Reporter Lysis buffer | Promega Biosciences Inc | Promega Biosciences Inc: E3971 | |
| Commercial assay or kit | X-treme Gene HP | Roche | Roche: 6366244001 | |
| Chemical compound, drug | KF116 | *Sharma et al., 2014* | | |
| Chemical compound, drug | BI224436 | MedChemExpress | MedChem Express: HY-18595 | |
| Chemical compound, drug | nicotinamide adenine dinucleotide phosphate (NADPH) | Sigma-Aldrich Chemical Company | Sigma-Aldrich: 10107824001 | |
| Chemical compound, drug | Verapamil Hydrochloride | Sigma-Aldrich Chemical Company | Sigma-Aldrich: V4629 | |
| Chemical compound, drug | Domperidone | Sigma-Aldrich Chemical Company | Sigma-Aldrich: D122 | |
| Chemical compound, drug | Chlorpromazine hydrochloride | Sigma-Aldrich Chemical Company | Sigma-Aldrich: C8138 | |
| Software, algorithm | HKL 3000 | HKL 3000 (http://hkl-xray.com) | RRID:SCR_015023 | |
| Software, algorithm | Phenix | Phenix (https://www.phenix-online.org/) | RRID:SCR_014224 | |
| Software, algorithm | Coot | Coot (http://www2.mrc-lmb.cam.ac.uk/personal/pemsley/coot) | RRID:SCR_014222 | |
| Software, algorithm | HADDOCK | *Dominguez et al., 2003*, *van Zundert et al., 2016* | | |
| Software, algorithm | ImageJ | ImageJ (https://imagej.net) | RRID:SCR_003070 | |
| Software, algorithm | OriginLab | OriginLab software (https://www.originlab.com) | | |
| Software, algorithm | ICY image analysis software | ICY image analysis software (https://icy.bioimageanalysis.org) | RRID:SCR_010587 | |

## Synthesis of ALLINIs

Racemic (+/-)-KF116 was synthesized as described previously (*Sharma et al., 2014*). Synthesis of (+) and (-) enantiomers of KF116 is detailed in the supporting information (*Supplementary file 1*). BI224436 was purchased from MedChemExpress.

## Protein purification

Hexa-His tagged recombinant WT full length IN, NTD-CCD, CCD and CCD-CTD domains were expressed in BL21 (DE3) *E. coli*. Full length IN and CCD-CTD were purified by nickel and heparin column as described previously (*Cherepanov, 2007*; *Kessl et al., 2012*). NTD-CCD and CCD domains were purified by nickel column (*Cherepanov, 2007*; *Kessl et al., 2012*) followed by SEC using HiLoad 16/600 Superdex column (GE healthcare) with the elution buffer of 20 mM HEPES (pH 7.5), 0.5M NaCl, 10% glycerol, 0.5 mM EDTA and 2 mM DTT at 0.5 mL/min flow rate. All the recombinant mutant proteins were generated by introducing substitutions into Hexa-His tagged pNL4-3 wild type IN by site directed mutagenesis and were purified similar to WT IN.

## SEC

Recombinant WT and mutant INs were analyzed on Superdex 200 10/300 GL column (GE Healthcare) with running buffer containing 20 mM HEPES (pH 7.5), 1 M NaCl, 10% glycerol and 5 mM BME at 0.3 mL/min flow rate. The proteins were diluted to 10 µM with the running buffer and incubated for 1 hr at 4°C followed by centrifugation at 10,000 g for 10 min. Different multimeric forms of IN were identified based on the following standards: bovine thyroglobulin (670,000 Da), bovine gamma-globulin (158,000 Da), chicken ovalbumin (44,000 Da), horse myoglobin (17,000 Da) and vitamin B12 (1,350 Da).

## DLS

The assays were performed on a Malvern Zetasizer Nano ZS as described previously (*Sharma et al., 2014*). IN domains were analyzed at 10 µM, whereas, full length WT and mutant INs were analyzed at 200 nM in the presence of 1 µM KF116 or BI224436. Kinetic analysis was carried out at specified time points.

## Inhibitor induced IN aggregation assay

5 µM of full length IN or NTD-CCD, CCD and CCD-CTD domains were incubated with increasing concentrations of KF116 and BI224436 in 20 µL reaction buffer containing 20 mM HEPES (pH 7.5), 1 M NaCl and 5 mM β-mercaptoethanol (BME). The reactions were incubated at 4°C overnight and centrifuged at 10,000*g for 10 min. The supernatant and the pellet fractions were analyzed by SDS-PAGE as described (*Hoyte et al., 2017*).

## Stoichiometry determination

Stoichiometry for KF116 and BI224436 induced IN aggregation was determined by adding increasing concentrations of inhibitors to 12.65 µM full length WT IN in 20 µL reaction buffer containing 20 mM HEPES (pH 7.5), 1 M NaCl and 5 BME. The reactions were incubated at 4°C overnight and centrifuged at 10,000*g for 10 min. Supernatant and pellet fractions were analyzed by SDS-PAGE as described (*Hoyte et al., 2017*). Aggregated IN from the pellet fraction was quantitated by ImageJ software. The data points were plotted by piecewise linear regression in OriginLab software and the stoichiometry for KF116 and BI224436 induced aggregation of IN were calculated as described (https://s3-us-west-2.amazonaws.com/oww-files-public/7/7b/FAQ_Stoichiometry_V03-2.pdf).

## X-ray crystallography

The HIV-1 IN CCD (residues 50–212) containing the F185H mutation was expressed and purified as described (*Sharma et al., 2014*). The protein was concentrated to 8 mg/ml and crystallized using hanging-drop vapor diffusion method with a crystallization buffer consisting of 100 mM sodium cacodylate pH 6.5, 100 mM ammonium sulfate, 10% (w/v) PEG 8000, and 5 mM DTT. Crystallization drops were prepared using an equal volume of protein and well solution. Crystallization trays were prepared on ice at room temperature and then transferred to 4°C for storage. Crystals formed within one week and continued to grow thereafter in size. Crystal data were collected on a Rigaku

Micromax-007 at 100 K. Data were integrated and scaled using HKL3000 (*Minor et al., 2006*) and Scalepack (*Otwinowski and Minor, 1997*). Phaser (*McCoy et al., 2007*) in the PHENIX suite (*Adams et al., 2010*) was used to run molecular replacement using Protein Data Bank code 4O55 as a search model (*Sharma et al., 2014*). Phenix.refine (*Afonine et al., 2012*) was used for data refinement, and manual refinement was done in Coot (*Emsley et al., 2010*). The coordinates are deposited in the Protein Data Bank under accession codes 6NUJ. The data and refinement statistics are given in *Supplementary file 1* (Table S1).

## Molecular docking studies

Inhibitor induced higher order HIV-1 IN multimer models were generated by the HADDOCK program (*Dominguez et al., 2003*; *van Zundert et al., 2016*) using IN tetramers and dimers as building blocks. To generate tetramer and dimer structures we used the cryo-EM structure of the HIV-1 intasome (*Passos et al., 2017*) and the crystal structure of the full-length HIV-1 IN containing ALLINI GSK1264 (*Gupta et al., 2016*), respectively. To include the ALLINI molecules in the starting structures, KF116 or BI224436 were docked into the canonical ALLINI binding sites based on the respective ligand binding mode seen in the crystal structures of KF116 (*Sharma et al., 2014*) and BI224436 (*Figure 6C*) bound to IN CCD dimer. Among the several docking poses generated by HADDOCK, the top poses that exhibited substantially higher buried surface area (BSA) compared with the rest of the poses were used for the analysis.

## Catalytic activities of mutant INs

Catalytic activities of WT and mutant INs were analyzed in the presence of LEDGF/p75 using homogeneous time-resolved fluorescence (HTRF) (*Slaughter et al., 2014*; *Tsiang et al., 2012*). HTRF offers an advantage of time resolved measurement of the energy transfer between donor and acceptor fluorophores, where a time delay between donor excitation and FRET measurement minimizes the background noise signal (*Degorce et al., 2009*). Briefly, 100 nM of each IN was incubated with 100 nM LEDGF/p75, 50 nM Cy-5 labeled donor DNA and 10 nM biotinylated target acceptor DNA in 20 mM HEPES (pH7.5), 1 mM DTT, 10 mM $MgCl_2$, 10% glycerol, 0.05% Brij-35 and 0.1 mg/ml BSA. End detection was based on europium-streptavidin antibody that binds to the biotinylated DNA and brings donor europium cryptate closer to acceptor Cy5 fluorophore in integrated DNA. This proximity results in energy transfer to yield a fluorescent signal that was recorded by Perkin-Elmer Life Sciences Enspire multimode plate reader. The data was plotted as percentage activity where the maximum HTRF signal from WT IN was set to 100% and the HTRF counts of different mutant INs were converted to relative percent activities.

## IN-LEDGF/p75 binding assay

Recombinant hexa-His tagged mutant INs were analyzed for their binding to LEDGF/p75 by affinity pull-down assay as described (*Hoyte et al., 2017*). Briefly, 1 μM of each IN was added to 1 μM of tag-less LEDGF/p75 in buffer containing 300 mM NaCl, 2 mM $MgCl_2$, 20 mM Imidazole, 0.2% (v/v) Nonidet P40, 50 mM HEPES(pH7.5) and 2 mM BME. The reaction mixture was incubated for 30 min at room temperature and added to the pre-equilibrated Nickel-Sepharose six fast flow beads (GE Healthcare). The bound fractions were later analyzed by SDS-PAGE and quantitated by ImageJ software.

## Virus production and reagents for live cell imaging

Plasmids encoding Vpr-INmNeonGreen (INmNG) and CypA-DsRed have been described previously (*Francis et al., 2016*; *Francis and Melikyan, 2018*). The HIV-1-based packaging vector pR9ΔEnv was from Dr. C. Aiken (Vanderbilt University). TZM-bl PPIA -/- cells depleted of CypA using CRISPR/Cas9 technology have been previously described (*Francis and Melikyan, 2018*). Fluorescently labeled pseudoviruses were produced and characterized, as described previously (*Francis et al., 2016*). Briefly, HEK293T/17 cells grown in 6-well culture plates were transfected with the following plasmids: HIV-1 pR9ΔEnv (2 μg), VSV-G (0.2 μg), Vpr-INmNeonGreen (0.5 μg) and, where indicated, CypA-DsRed (0.5 μg) using the JetPrime Transfection reagent (VWR, Radnor, PA). Six hours after transfection, the medium was replaced with fresh DMEM/10% FBS without phenol red, and the cells were cultured for additional 36 hr at 37°C, 5% $CO_2$. Viral supernatant was

collected, filtered through a 0.45 µm filter, aliquoted and stored at −80℃ until use. Where indicated, virus production was carried out in the presence of 5 µM of (+/-)-KF116, (-)-KF116, BI224436 or RAL.

## Fluorescence imaging and analysis

TZM-bl PPIA-/- cells ($5 \times 10^5$) were grown on 35 mm glass-bottom dishes (MatTek Corp.). Tracking of single HIV-1 particles pseudotyped with VSV-G and co-labeled with INmNG/CypA-DsRed in live cells was performed, as previously described (*Francis et al., 2016*). In brief, cell nuclei were stained for 10 min with 2 µg/ml Hoechst-33342 followed by spinoculation with pseudoviruses (10 pg of p24 per $5 \times 10^5$ TZM-bl PPIA-/- cells, MOI 0.008) at $1500 \times$ g, 4℃ for 30 min. The cells were washed twice, and virus entry was initiated on a temperature- and $CO_2$-controlled microscope stage by adding pre-warmed live-cell imaging buffer (Life Technologies, Invitrogen) supplemented with 10% FBS. 3D time-lapse imaging was carried out with a Zeiss LSM880 AiryScan confocal microscope using a C-Apo 63x/1.4NA oil-immersion objective. A suitable field of view was selected, and full cell volume was imaged every 10 s using confocal or AiryScan fast mode by acquiring 11–17 Z-stacks spaced by 0.5 µm, using a low power of 405, 488 and 561 nm lasers for Hoechst-33342, NeonGreen and DsRed, respectively. A DefiniteFocus module (Carl Zeiss) was utilized to correct for axial drift during image acquisition.

Fixed time-point imaging was performed using AiryScan Optimal mode at room temperature using sequential imaging of multiple fields of view, each typically containing ~7 cells. After acquisition, the images was processed by applying auto-threshold parameters. Acquired image series were converted to maximum intensity projections and analyzed using the ICY image analysis software (icy.bioimageanalysis.org). Single particle tracking was performed using INmNG as a reference channel for DMSO treated control samples and CypA-DsRed as reference for ALLINI treated infections to demonstrate uncoating and INmNG signal disappearance, respectively. Single virus fluorescence and colocalization analysis was performed using the ROI detector and Colocalization studio plugin from ICY software. Single particle intensity analyses was performed after background subtraction and by normalizing to the initial fluorescence intensity to 100%.

## Viral plasmids and cloning

The N155H/K156N and N155H/K156N/K211R/E212T substitutions were cloned into the IN coding region of pNL4-3.Luc.E⁻R⁺ (*Connor et al., 1995*) using PCR-site directed mutagenesis (Agilent) and verified by dideoxy sequencing. Additional IN substitutions at K14A, Y15A, T210 +Pro and N222K were similarly introduced into pNL4.3 (*Adachi et al., 1986*).

## Cells, transfections, and infections

HeLa and HEK293T cells were purchased from ATCC, while TZM-bl and MT-4 cells were obtained from NIH AIDS reagent program. The identity of all cell lines have been authenticated by STR profiling done at ATCC. Cell lines were tested at periodic intervals for *Mycoplasma* contamination by using Mycoscope Mycoplasm PCR detection kit and there has been no evidence of *Mycoplasma*. HEK293T cells were cultured in Dulbecco's modified eagle medium (Gibco), 10% FBS (Sigma-Aldrich) and 1% Penicillin Streptomycin (Gibco) at 37℃ and 5% $CO_2$. MT-4 cells were cultured in Roswell Park Memorial Institute (RPMI) 1640 medium (Gibco) supplemented with 10% FBS and 1% Penicillin Streptomycin. Virus stocks were produced as previously described (*Sharma et al., 2014*).

## Infectivity of the mutant viruses

For viral infectivity experiments, virions produced from HIV-1$_{NL4-3}$, HIV-1$_{NL4-3}$ IN (K14A), HIV-1$_{NL4-3}$ IN (Y15A), HIV-1$_{NL4-3}$ IN (T210 +Pro) and HIV-1$_{NL4-3}$ IN (N222K) molecular clones were generated by transfecting HEK293T cells as previously described (*Feng et al., 2013*). Twenty-four hours post-transfection, the culture supernatant was replaced with fresh complete medium after washing once with complete medium. Forty-eight hours post-transfection, the virus containing supernatant was collected, and filtered through 0.45 µm filter. TZM-bl cells ($2 \times 10^5$) were then infected with HIV-1 virions equivalent to 100 ng of virus determined by HIV-1 Gag p24 ELISA (Zeptometrix) following manufacturer's protocol in the presence of 8 µg/ml Polybrene (Sigma). Two hours post-infection the culture supernatant was removed, and the cells were washed and replaced with fresh medium. Viral infectivity was determined using the procedure previously described (*Feng et al., 2013*).

## Antiviral assays

$EC_{50}$ values of (+) and (-) enantiomers of KF116 against replication competent HIV-1$_{NL4-3}$ were assayed in MT-4 cells. 50 µl of 2x test concentration of the diluted compounds were added to RPMI 1640 cell culture medium with 10% FBS added to each well of a 96-well plate (nine concentrations) in triplicate. MT-4 cells were infected with HIV-1$_{NL4-3}$ at a multiplicity of infection (MOI) of 0.01 for 3 hr. Fifty microliters of infected cell suspension in culture medium with 10% FBS ($\sim$1.5$\times$10$^4$ cells) were then added to each well containing 50 µl of diluted compound. The plates were then incubated at 37°C 5% $CO_2$ for 5 days. After 5 days of incubation, 100 µl of CellTiter-Glo reagent (catalog no. G7571; Promega Biosciences, Inc) was added to each well containing MT-4 cells. Cell lysis was carried out by incubation at room temperature for 10 min, and chemiluminescence was read.

The $EC_{50}$ values of (-)-KF116 and BI224436 against WT, N155H/K156N and N155H/K156N/ K211R/E212T viruses were determined in single replication cycle as described previously (*Sharma et al., 2014*). Briefly, HEK293T cells were transfected with WT and mutant pNL4-3.Luc.Env- and pCG-VSV-G (*Brown et al., 1999*) plasmids to produce respective viruses in the absence and presence of indicated concentrations of ALLINIs. The virus supernatants were collected 24 hr after drug treatment and p24 concentrations were determined by HIV-1 Gag p24 ELISA (ZeptoMetrix) following manufacturer's protocol. The target HeLa cells were then infected with HIV-1 virions equivalent to 10 ng of HIV-1 p24, in the presence of indicated concentrations of drugs. Two hours post-infection the culture supernatant was removed, washed once with complete medium, and then fresh complete medium was added with the inhibitor concentration maintained. The cells were cultured for 48 hr and the cell extracts were prepared using reporter lysis buffer (Promega). Luciferase activity was determined using a commercially available kit (Promega). The fitted dose-response curves were generated to calculate $EC_{50}$ using OriginLab software.

## ADME studies

The metabolic stability of (+/-)-KF116, (-)-KF116 and BI224436 was evaluated in rat and human liver microsomal incubations by probing in vitro Cytochrome (CYP) P450 activity in the presence of co-factor NADPH. Stock DMSO solutions were prepared and LC/MS-MS methods were developed. Verapamil, domperidone and chlorpromazine were used as controls. Liver microsomal incubations (male Sprague-Dawley rat, p=500, 0.2 mg protein/mL; and human mixed gender, p=50, 1.0 mg protein/mL) were performed in the presence of NADPH (1.0 mM), taking samples as a function of time and analyzing drug concentration (*Wempe and Anderson, 2011*; *Wempe et al., 2012a*; *Wempe et al., 2012b*). In this assay, the loss of parent drug is monitored over-time to produce in vitro half-life values. These half-life results were used to calculate intrinsic clearance values ($CL_{int}$, reported in units of µL/min/mg protein), as previously described using the following equation: (0.693/in vitro T1/2) • (µL incubation/mg microsomal protein) (*Obach, 1999*).

## Acknowledgements

We thank Dr. Jared M Brown for providing us with access to the DLS instrument at the University of Colorado Anschutz Medical Campus. We also thank Jessica Bruhn for critical reading of the manuscript and helpful suggestions.

## Additional information

### Competing interests

Alan N Engelman: ANE reports fees from ViiV Healthcare Co. outside of the current work. The other authors declare that no competing interests exist.

### Funding

| Funder | Grant reference number | Author |
| --- | --- | --- |
| National Institutes of Health | U54GM103368 | Mamuka Kvaratskhelia |
| National Institutes of Health | R01AI062520 | Mamuka Kvaratskhelia |

| National Institutes of Health | KL2 TR001068 | Ross C Larue |
|---|---|---|
| National Institutes of Health | R37AI039394 | Alan N Engelman |
| National Institutes of Health | R01AI143649 | Mamuka Kvaratskhelia |
| National Institutes of Health | R01AI129862 | Gregory B Melikyan |

The funders had no role in study design, data collection and interpretation, or the decision to submit the work for publication. The content is solely the responsibility of the authors and does not necessarily represent the official views of the National Institutes of Health.

## Author contributions

Pratibha C Koneru, Conceptualization, Data curation, Formal analysis, Investigation, Methodology, Writing—original draft, Writing—review and editing; Ashwanth C Francis, Nanjie Deng, Stephanie V Rebensburg, Ashley C Hoyte, Jared Lindenberger, Daniel Adu-Ampratwum, Ross C Larue, Michael F Wempe, Formal analysis, Investigation, Methodology, Writing—review and editing; Alan N Engelman, Dmitry Lyumkis, James R Fuchs, Ronald M Levy, Gregory B Melikyan, Conceptualization, Writing—review and editing; Mamuka Kvaratskhelia, Conceptualization, Resources, Data curation, Supervision, Funding acquisition, Methodology, Writing—original draft, Project administration, Writing—review and editing

## Author ORCIDs

Pratibha C Koneru https://orcid.org/0000-0003-3955-4548
Daniel Adu-Ampratwum http://orcid.org/0000-0001-9392-2431
Dmitry Lyumkis http://orcid.org/0000-0002-8124-7472
Ronald M Levy http://orcid.org/0000-0001-8696-5177
Mamuka Kvaratskhelia https://orcid.org/0000-0003-3800-0033

## Decision letter and Author response

Decision letter https://doi.org/10.7554/eLife.46344.sa1
Author response https://doi.org/10.7554/eLife.46344.sa2

# Additional files

## Supplementary files

- Supplementary file 1. Synthesis of (+) and (-) enantiomers of KF116.
- Transparent reporting form

## Data availability

Diffraction data have been deposited in PDB under the accession code 6NUJ.

The following dataset was generated:

| Author(s) | Year | Dataset title | Dataset URL | Database and Identifier |
|---|---|---|---|---|
| Lindenberger JJ, Kvaratskhelia M | 2019 | HIV-1 Integrase Catalytic Core Domain Complexed with Allosteric Inhibitor BI-224436 | http://www.rcsb.org/structure/6NUJ | RCSB Protein Data Bank, 6NUJ |

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
