## [Decision Letter]

Thank you for submitting your article "HIV-1 Integrase Tetramers are the Antiviral Target of Pyridine-based Allosteric Integrase Inhibitors" for consideration by *eLife*. Your article has been reviewed by three peer reviewers, including Julie Overbaugh as the Reviewing Editor and Reviewer #1, and the evaluation has been overseen by Arup Chakraborty as the Senior Editor. The following individuals involved in review of your submission have agreed to reveal their identity: Eric Freed (Reviewer #2).

The reviewers have discussed the reviews with one another and the Reviewing Editor has drafted this decision to help you prepare a revised submission.

The studies of Konerua focus on the mechanism of action of ALLINI-based inhibitors and specifically the form of IN they target. The studies are comprehensive and paint a consistent picture of ALLINI interaction with the IN tetramer. They reveal a new model for ALLINI-based inhibitors that refines and clarifies some prior studies and shows that these compounds preferentially bind to tetrameric from of IN. They authors support this conclusion with a variety of different experimental approaches. The studies also reveal similarities and differences for quinoline and pyridine based ALLINI-IN inhibitors. Finally, the studies suggest a role for the NTD, working in an indirect manner to promote Inhibitor-IN interactions leading to multimerization.

Summary:

In this study, Koneru and colleagues investigated the target and mechanism of action of allosteric HIV-1 integrase (IN) inhibitors (ALLINIs). These compounds are distinct from IN strand-transfer inhibitors (INSTIs) in that they act (predominantly) during the maturation step of the replication cycle by inducing premature aggregation of IN, thereby disrupting proper condensation of the viral genomic RNA within the nascent capsid. The authors provide data demonstrating that the pyridine-based ALLINI KF116 selectively targets IN tetramers, at least in vitro. They synthesize a highly active enantiomer of KF116 and use live-cell imaging to show that, early post-infection, virions formed in the presence of KF116 lose the tight association between capsids and IN that is observed in untreated virions. As expected, based on its distinct mechanism of action, KF116 retains activity against a DTG-resistant IN mutant.

The ALLINIs are an important class of novel anti-HIV compounds that may turn out to be clinically viable, and studies performed with these inhibitors (many of which were from these authors) have provided key insights into the role of IN in HIV-1 replication beyond the integration step itself. The current work is of high quality and provides important new insights into ALLINI activity.

The observation that the enantiomer (-)-KF116 is significantly better than the (+)- KF116 for high potency inhibition of virus replication and metabolic stability is exciting and promising. The time experiment showing the association of IN with the viral CA in virus infected cells, in the presence or absence of both ALLINIs, is highly suggestive of their association in vivo and supports their central mechanism for inhibiting viral replication by ALLINI. Their controls are highly appropriate.

Essential revisions:

Most comments are focused on making the presentation of the data more accessible to a broad audience. In many instances, the authors present data without sufficient explanation of how experiments were performed and how the results should be interpreted.

1) Figure 1C. The authors should explain that only high-MW aggregates are detectable using dynamic light scattering (e.g., IN monomers, dimers, and tetramers are not detected).

2) Figure 4, Figure 5 and associated supplementary Figures. The authors construct a series of IN mutations and examine their effect on the ability of KF116 and the BI compound to induce higher-order aggregation. Based on the data, the authors argue that the IN N-terminal domain (NTD) and CCD-CTD linker are critically important for ALLINI-induced IN multimerization. However, there is a perfect correspondence between the effect of the mutations on IN tetramer formation and the ability of KF116 to induce higher-order multimerization. It thus appears that the mutations may not be directly affecting KF116-induced aggregation, but rather are blocking IN tetramerization, which is required for KF116 activity. This should be explained more clearly.

3) Figure 8 and associated supplementary Figures make a nice addition to the paper, albeit somewhat off the main focus of the study. However, these figures may be somewhat difficult for readers who are not familiar with the HIV uncoating and live-cell imaging literature. In particular, the presentation of Figure 8—figure supplement 2 could be improved. There is only one sentence in the text devoted to this figure. What is the significance of using the two different microscopes in panel A vs B? At a glance, it appears that the graphs represent quantification of the images, yet the time scales are very different. The images in A extend out to 6 minutes 23 seconds, yet the graph goes out to >60 minutes. What information does this figure provide beyond what is shown in previous Figure 8 panels?

4) Since IN-RNA interactions allow packaging of both into the core capsid (Madison, et al., 2017), would these inhibitors in the presence of IN-viral RNA complex have the same multimerization effects as described above for purified wt HIV-1 IN by itself? A brief discussion of this possibly for investigators not familiar with this field appears appropriate in this manuscript, enough though past studies suggest that IN is separated from viral RNA in virions upon ALLINI treatment.

---

## [Author Response]

Essential revisions:Most comments are focused on making the presentation of the data more accessible to a broad audience. In many instances, the authors present data without sufficient explanation of how experiments were performed and how the results should be interpreted.1) Figure 1C. The authors should explain that only high-MW aggregates are detectable using dynamic light scattering (e.g., IN monomers, dimers, and tetramers are not detected).

We have added the following explanation in subsection “KF116 specifically and BI224436 preferentially targets full-length WT IN tetramers”: “DLS is an optical method for studying the diffusion behavior of macromolecules in solution (Stetefeld et al., 2016). While unliganded IN does not yield any detectable signal due to relatively small sizes of fully soluble monomeric, dimeric and tetrameric forms, ALLINI induced higher-order IN oligomers are readily detected by DLS (Sharma et al., 2014).”

2) Figure 4, Figure 5 and associated supplementary Figures. The authors construct a series of IN mutations and examine their effect on the ability of KF116 and the BI compound to induce higher-order aggregation. Based on the data, the authors argue that the IN N-terminal domain (NTD) and CCD-CTD linker are critically important for ALLINI-induced IN multimerization. However, there is a perfect correspondence between the effect of the mutations on IN tetramer formation and the ability of KF116 to induce higher-order multimerization. It thus appears that the mutations may not be directly affecting KF116-induced aggregation, but rather are blocking IN tetramerization, which is required for KF116 activity. This should be explained more clearly.

We fully agree with the reviewers’ interpretation. Indeed, our goal was to target select NTD and CCDCTD linker amino acids to specifically compromise IN tetramerization without directly affecting ALLINI binding sites. As the reviewers correctly noted we observed a prefect correspondence between effects on IN tetramerization and KF116 induced higher-order IN oligomerization, which is a key finding of the manuscript. To make this point abundantly clear we have revised the text in subsection “NTD and the α-helical linker connecting CCD with CTD are critically important for ALLINI induced higher-order IN multimerization”.

The following sentence was added to subsection “NTD and the α-helical linker connecting CCD with CTD are critically important for ALLINI induced higher-order IN multimerization”, which further explains the rationale for the site-directed mutagenesis experiments: “Therefore, to probe the significance of IN tetramerization for ALLINI induced higher-order IN multimerization we targeted these protein regions (the NTD and the CCD-CTD linker) with site directed mutagenesis to specifically compromise functional IN tetramerization without affecting the direct sites of ALLINI binding.”

In addition, we have revised the concluding paragraph for subsection “NTD and the α-helical linker connecting CCD with CTD are critically important for ALLINI induced higher-order IN multimerization”as follows: “Collectively, the experiments with select substitutions in the NTD and CCD-CTD linker, which allowed us to compromise functional IN tetramerization without directly affecting ALLINI binding sites, revealed a consistent correspondence between the effects of the substitutions on IN tetramer formation and the ability of KF116 to induce higher-order multimerization.”

3) Figure 8 and associated supplementary Figures make a nice addition to the paper, albeit somewhat off the main focus of the study. However, these figures may be somewhat difficult for readers who are not familiar with the HIV uncoating and live-cell imaging literature. In particular, the presentation of Figure 8—figure supplement 2 could be improved. There is only one sentence in the text devoted to this figure.

We have extensively revised the text in the subsection “ALLINI treatment compromises the association of IN with the viral core” and included schematic depictions in new Figure 8—figure supplement 2, which will help the general reader to better follow our experimental design and results.

What is the significance of using the two different microscopes in panel A vs B?

Imaging was performed on the same LSM880 AiryScan microscope using both confocal and AiryScan modes, and the results were not affected by imaging mode. The AiryScan imaging allows for sensitive detection of viral complexes with high temporal resolution and reduces photobleaching. To avoid confusion, we removed the images acquired in a regular confocal imaging mode. Furthermore, we have replaced previous Figure 8—figure supplement 2, which was redundant with main Figure 8, and instead included a new illustrative Figure 8—figure supplement 2 to summarize our experimental findings of INmNG and/or CypA-DsRed signal loss during productive (in the absence of the inhibitor) vs non-productive (in the presence of ALLINI) HIV-1 infection.

At a glance, it appears that the graphs represent quantification of the images, yet the time scales are very different. The images in A extend out to 6 minutes 23 seconds, yet the graph goes out to >60 minutes. What information does this figure provide beyond what is shown in previous Figure 8 panels?

We apologize for the confusion related to the graph. In fact, we agree with the reviewers that main Figure 8 sufficiently depicts our experimental findings and that additional experimental results included in previous Figure 8—figure supplement 2 were redundant. Therefore, we have replaced the previous Figure 8—figure supplement 2 with schematic summary (see new Figure 8—figure supplement 2) that will make our manuscript more amenable to the general reader.

4) Since IN-RNA interactions allow packaging of both into the core capsid (Madison, et al., 2017), would these inhibitors in the presence of IN-viral RNA complex have the same multimerization effects as described above for purified wt HIV-1 IN by itself? A brief discussion of this possibly for investigators not familiar with this field appears appropriate in this manuscript, enough though past studies suggest that IN is separated from viral RNA in virions upon ALLINI treatment.

We have included a new schematic Figure 8—figure supplement 2, which summarizes our experimental findings (A) and compares untreated vs ALLINI-treated virions (B). In addition, we have added the respective text in the Discussion section: “The schematic summary of these experimental observations are depicted in Figure 8—figure supplement 2A. In turn, the pre-mature loss of IN from the viral CA cores during infection of target cells is likely to be a consequence of aberrant packaging of ALLINI induced higher order IN multimers within virions (Figure 8—figure supplement 2B). Collectively, our current findings together with published results (Kessl et al., 2016) suggest that binding of functional IN tetramers to the viral RNA genome are important for effective packaging of these viral components within conical CA cores and formation of the infectious virions (Figure 8—figure supplement 2). In contrast, ALLINI treatment results in aggregation and separation of IN from RNPs. As a result, both IN-ALLINI aggregates and RNPs lacking IN are mislocalized outside of the protective CA cores and yield eccentric, non-infectious virions (Figure 8—figure supplement 2B).”

We believe that these changes collectively will benefit general readers.